# Analgesia for Sheep in Commercial Production: Where to Next?

**DOI:** 10.3390/ani11041127

**Published:** 2021-04-14

**Authors:** Alison Small, Andrew David Fisher, Caroline Lee, Ian Colditz

**Affiliations:** 1CSIRO Agriculture & Food, Locked Bag 1, Armidale, NSW 2350, Australia; caroline.lee@csiro.au (C.L.); ian.colditz@csiro.au (I.C.); 2Animal Welfare Science Centre, University of Melbourne, Parkville, VIC 3052, Australia; adfisher@unimelb.edu.au

**Keywords:** animal welfare, husbandry procedure, pain, NSAID, castration, tail docking, mulesing, lamb, anaesthesia

## Abstract

**Simple Summary:**

Increasing societal and customer pressure to provide animals with ‘a life worth living’ continues to apply pressure on industry to alleviate pain associated with husbandry practices, injury and illness. Although a number of analgesic solutions are now available for sheep, providing some amelioration of the acute pain responses, this review has highlighted a number of potential areas for further research.

**Abstract:**

Increasing societal and customer pressure to provide animals with ‘a life worth living’ continues to apply pressure on livestock production industries to alleviate pain associated with husbandry practices, injury and illness. Over the past 15–20 years, there has been considerable research effort to understand and develop mitigation strategies for painful husbandry procedures in sheep, leading to the successful launch of analgesic approaches specific to sheep in a number of countries. However, even with multi-modal approaches to analgesia, using both local anaesthetic and non-steroidal anti-inflammatory drugs (NSAID), pain is not obliterated, and the challenge of pain mitigation and phasing out of painful husbandry practices remains. It is timely to review and reflect on progress to date in order to strategically focus on the most important challenges, and the avenues which offer the greatest potential to be incorporated into industry practice in a process of continuous improvement. A structured, systematic literature search was carried out, incorporating peer-reviewed scientific literature in the period 2000–2019. An enormous volume of research is underway, testament to the fact that we have not solved the pain and analgesia challenge for any species, including our own. This review has highlighted a number of potential areas for further research.

## 1. Introduction

Increasing societal and customer pressure to provide animals with ‘a life worth living’ continues to apply pressure on industry to alleviate pain associated with husbandry practices, injury and illness. A substantial body of research into the impacts of sheep husbandry procedures such as castration, tail docking and mulesing has highlighted the need for analgesic strategies or alternative management approaches, and in recent years non-steroidal anti-inflammatory drugs (NSAIDs) and local anaesthetic formulations specifically registered (authorised by the Competent Authority, also known as ‘licensed’ in some jurisdictions) and delivery methodologies specifically designed for sheep have become available in a number of countries. However, even with multi-modal approaches to analgesia, using both local anaesthetic and NSAIDs, pain is not obliterated, and the challenge of pain mitigation remains. It is timely to review and reflect on progress to date in order to strategically focus on the most important challenges, and the avenues which offer the greatest potential to be incorporated into industry practice in a process of continuous improvement.

This review aimed to provide a stocktake of recently published (in the years 2000–2019) research into pain mitigation for livestock and other research into pain and analgesia whose findings are potentially relevant to the Australian sheep industry.

## 2. Materials and Methods

A structured, systematic approach to the literature search was developed, according to the criteria outlined in Table 1, and using the keywords listed in Table 2.

Following an initial screen of search results (title and publication date) against the inclusion and exclusion criteria, the bibliographical data for remaining articles were imported to an Endnote™ database, and efforts were made to access the full text of each article.

Each article was then read in detail, extracting the information listed in Table 3, and an evaluation of the existing knowledge and gaps in knowledge was made.

## 3. Results and Discussion

### 3.1. Outcomes of Literature Search

Initial searches returned over 100,000 potential articles for assessment. Where the initial search generated a very large number of articles related to detailed investigation of the physiological minutiae of a particular pharmacological approach, a subset of papers relating particularly to use in livestock were retained, and the remaining articles excluded from this review (for example, the single keyword ‘analgesia’ generates over 74,000 articles published between 2000 and 2019 when entered into Web of Science^®^). The enormous volume of research underway is testament to the fact that we have not solved the pain and analgesia challenge for any species, including our own. Due to this overwhelming number of potential inclusions, the literature search was carried out in an iterative manner, focusing on a particular topic subsection at a time, and further articles were located during the course of preparation of this review. Following initial screening of title and abstract, 1305 articles were selected for further evaluation, and bibliographic data for each of these was imported into an Endnote™ database.

Subsequently, 338 articles were excluded as either being of a review or perspectives nature; in a language other than English; published prior to 2000; or not containing any information that would be currently useful to the sheep industry, and detailed information was collated for 967 articles. Some review articles are mentioned in this review, as is some literature published prior to 2000, in order to provide context to the discussion.

### 3.2. Pain

Pain is an aversive sensory and emotional experience, leading to changes in physiology and behaviour [1]. At the most basic level, nociception, peripheral receptors detect a noxious stimulus (transduction), that information is transmitted to the spinal cord via nerve fibres (transmission), the information is then modulated and projected to the brain, leading to the ultimate perception of the stimulus by the individual [2]. Perception of pain initiates behavioural and physiological actions as the animal attempts to avoid the noxious stimulus or control its impact. Thus, pain is a welfare concern both for the negative emotional experience it generates and for the adverse behavioural and physiological consequences it causes. There is a large variety of peripheral receptors, designed to detect a variety of stimuli. Pain mitigation strategies aim to reduce both the emotional perception of pain and the activation of adverse behavioural and physiological responses. Analgesic agents achieve this by targeting particular receptors or particular biochemical cascades, which may be at the peripheral or central level, or both, aiming to reduce the scale of the noxious input to the brain. Some agents also act to limit physiological and immunological (inflammatory) responses via non-neuronal pathways. To date, the majority of the research into analgesia for livestock species has focused on the acute or immediate pain associated with husbandry procedures. However, there is increasing understanding that post-procedural pain may persist for more than a day or two, and it is likely that chronic or neuropathic pain may also be experienced [3,4,5,6,7,8]. Neuropathic pain results from neuronal plasticity (re-wiring) after nerve damage. Nerve damage can occur through injury, surgery, metabolic disruptions or viral and other infections. Challenges facing the introduction of treatment of neuropathic pain in livestock include the chronic nature of neuropathic pain, requiring ongoing therapy (unless prevention of development of neuropathic pain can be achieved); and the fact that understanding of the physiology and development of therapeutic approaches to neuropathic pain in humans is also still very much in its infancy.

A detailed discussion of the physiology of pain in mammals is beyond the scope of this review, although some detail will be given against specific analgesic strategies or husbandry procedures where appropriate. For further detail on pain physiology, a number of comprehensive reviews are available, for example Frias and Merighi 2016 [9].

#### 3.2.1. Non-Pharmacological Modulators of the Pain Response

A variety of factors influence the pain response. There may be gender-specific effects, particularly in the neonatal phase when the nervous system is still maturing, the sensitivity to a thermal stimulus reducing over the first 12 days of ex utero life in male lambs, whereas in female lambs there was no change in sensitivity [10]. In general, however, gender does not significantly affect pain responses whereas age or bodyweight can [10,11]. There was an increasing intensity of electroencephalographic (EEG) changes in response to castration as lambs increased in age from 1 day to 36 days [12], and similarly, the EEG responses to tail docking were significantly more marked in piglets docked at 20 days of age as compared to 2 days of age [13]. A further study identified that male lambs castrated at one day of age had a greater pain response to tail docking at 3–5 weeks of age than lambs that were castrated at 10 days of age [14], i.e., an insult during the early neonatal development of the nervous system led to increased pain sensitivity later in life. Interestingly, all three of these studies used male animals: in the context of the findings of Guesgen et al. [10], would the results have differed if female lambs were studied? Psychosocial factors can also affect animal behaviour, leading to social contagion or social buffering, and the behavioural response to castration or tail docking can be reduced by the presence of a familiar conspecific [15,16]. Familiarity with the environment and familiarity with being handled can also reduce the behavioural response to a painful procedure such as tail docking [16,17].

#### 3.2.2. Section Summary

Detailed understanding of the physiology of pain perception, particularly at the molecular level, is still developing. There is increasing evidence that the pain perception pathways continue to develop after birth, and events in the early neonatal period can affect subsequent pain sensitivity. A number of non-pharmacological factors can affect the pain response, and these warrant further investigation toward the development of an holistic approach to integrated pain management.

### 3.3. Pain Assessment

Physiological and behavioural changes are rarely pathognomonic of pain. This means that the context within which changes are seen can influence their interpretation as being indicative of the presence or severity of pain. As a consequence, the severity of pain and strategies for its mitigation are best examined in carefully conducted studies where putative measures are assessed in treatment groups including positive and negative control treatments. Furthermore, as in humans, the perception and expression of pain differ between individual animals. This could occur due to genetic and non-genetic differences between individuals anywhere along the afferent (e.g., nociceptor function), central, or efferent (e.g., behavioural) arms of pain responses. For instance, temperament has been shown in horses and dogs to be one such factor differing between individuals that can influence expression of pain behaviours [18,19]. Age, prior experience, and current affective state can also influence an individual’s perception of pain and the expression of efferent changes indicative of pain. Differences between individuals in responses to a standardised test paradigm such as ring castration create a dilemma for animal welfare science. The ethics of animal welfare dictate that the experience of the individual animal matters, yet at present our capacity to describe the experience of the individual animal is limited [20]. Rather, studies contrasting changes between various treatment groups lead to the description of the *average* types and magnitudes of changes indicative of pain rather than providing measures finely calibrated to the particularities of the individual. A discussion of some general types of changes that are sometimes indicative of pain follows. The cascade of efferent changes initiated by central perception of pain includes a suite of changes in physiological and behavioural activities. A number of these changes attract attention as indicators of pain. Variation in the physiological and behavioural responses seen between individuals and between causes of pain has stimulated the search for a central (mind based) “gold standard” of the experience of pain. This quest has led to research on the affective state of the animal (as the central barometer of the experience of pain). Once again, a dilemma arises as the affective state of the animal currently needs to be imputed indirectly through its influence on expressed functions such as cognition, demeanour and activity. Additionally, yet again, factors unrelated to pain (or affective state) can influence performance of the individual in these tests. Some of the commoner measures and some promising newly emerging measures are described below.

#### 3.3.1. Physiological Indictors

Oxidative stress measures appear to alter with pain, as demonstrated in cattle undergoing dehorning, by changes in thiobarbituric acid-malondialdehyde (TBA-MDA); nitric oxide (NO); plasma antioxidant activity (AOA); glutathione (GSH); cortisol; glucose; cholesterol. MDA, NO, AOA and GSH were the most powerful markers for evaluating the oxidant/antioxidant status in mature cattle [21]. While measures of oxidative stress have not commonly been used in pain assessment, these measures appear promising and should be explored in future studies.

Physiological measures of blood pressure (systolic, diastolic and mean arterial BP) and heart rate (HR) may be more sensitive than adrenocorticotrophic hormone (ACTH) or cortisol as indices of low-grade pain or persisting sympathetic tone in sheep. One hour after lambs were castrated and tail docked, there was an increase in BP, HR, ACTH and cortisol. However, by 4 h post-treatment, both ACTH and cortisol had returned to baseline and both BP and HR were still elevated [22].

Use of infrared thermal imaging (IRT) to measure eye temperature shows promise as a non-invasive measure of pain. Using a castration model in cattle, Stewart et al. [23] found that eye temp and cortisol increased in surgical castration groups, and this was less so with local anaesthetic. Heart rate variability (HRV) was shown to be more sensitive than eye temperature using IRT for assessing mild to moderate ischaemic pain in sheep [24]. However, HRV is more difficult to measure in practical settings and requires complex analytic software. Stubsjoen et al. [25] further investigated heart rate measures looking at detrended fluctuation analysis (DFA): fractal analysis of heart rate variability at baseline, intervention (ischaemic pain) and post-intervention and found that DFA of heart rate time series could help to evaluate sympathetic activation and vagal withdrawal due to pain. In a footrot pain model in sheep, HRV was effective at measuring clinical disease [26].

Nociceptive threshold has been used in many species with varying results. Nociceptive stimuli can be thermal (hot or cold), electrical impulse, pin-prick, crushing/pinching or pressure, including use of Von Frey filaments or Algometer pressure devices [11,27,28,29,30,31,32,33,34,35,36,37,38,39,40,41,42,43,44,45,46,47,48,49,50,51,52,53,54,55,56,57,58,59,60,61,62,63]. The measure is rapid and practical to apply, especially with the trialling of remote methods for sheep [64]. In pigs, application of pressure measurement to the tail was effective but younger pigs had higher sensitivity, lower thresholds and lower variability than older animals [32]. Age and bodyweight were also reported to affect the algometer response in pigs [11]. Additionally, lameness results in hyperalgesia (increased sensitivity) that is not necessarily pain related in pigs [65]. In more stoic species, such as the donkey, nociceptive thresholds are more challenging to assess as no response may be elicited [39]. In lambs, nociceptive thresholds were higher when individuals were more closely following their mothers, indicating that they were less sensitive to or demonstrative of pain [41]. These factors should be considered when applying nociceptive threshold measures, using similar animals in terms of age and considering species specific behaviour will reduce variation due to these effects. Overall, nociceptive thresholds are a well-established measure of pain sensitivity and relatively easy to conduct.

Electroencephalography (EEG) measures have been used to compare responses between surgical husbandry procedures in sheep but these are impractical to apply due to the restraint needed [66]. However, more recently, implanted EEG devices have been applied that enable brain measurement in unrestrained sheep [67]. Although these show promise for research purposes, they are likely to be more useful in small-scale controlled research applications that in large-scale commercial monitoring.

#### 3.3.2. Biomarkers

Immune factors and inflammatory cytokines were recommended in a systematic review by Hazel et al. [68] as potential novel biomarkers of pain. Pang et al. [69] found no changes in inflammatory cytokines with castration in cattle, whereas in pigs with lameness and rectal prolapse, salivary biomarkers (cortisol, salivary alpha-amylase, total esterase activity, butyrylcholinesterase, adenosine deaminase isozymes (ADA1) and (ADA2) all showed differences compared to control animals [70]. Salivary measures are non-invasive and may be relatively practical to use in livestock studies. 

Acute phase protein serum amyloid was an effective measure of inflammation in horses undergoing standing surgical castration [71]. Cortisol has been widely used in many studies. However, care must be taken over interpretation, as a cortisol response can be elicited by any form of arousal, not merely pain.

#### 3.3.3. Behaviours

Individual behaviours and ethograms are good measures when comparing analgesic efficacy between treatment groups receiving the same husbandry procedure. Well-demonstrated effects of pain on sheep include changes in behavioural time budgets such as reduced lying and increased abnormal postures, plus reduced playing activity in response to procedures such as castration [72]. Similarly, for cattle, lying time is reduced in response to castration [73].

Individual behaviours and ethograms can differ between procedures. In sheep and cattle for instance, ring and knife castration evoke different behavioural changes that make comparison of the relative painfulness and efficacy of pain mitigation strategies between the procedures difficult [74]. As well, there is evidence in mice that strain specific behaviours occur following vasectomy [75], which may have implications for comparing behaviours across different breeds, or genetic lines, of sheep.

Low-frequency behaviours such as drinking and the suite of behaviours interpreted as indicating the lamb is experiencing pain usually occur at too low a frequency for statistical analysis of the individual behaviours. As a consequence, it is standard practice to sum the pain associated behaviours into a single class of “abnormal behaviours” for statistical analysis. This is a robust methodology for comparing a husbandry procedure performed with and without analgesic drugs. In that scenario, the husbandry procedure under examination is fixed and the question is whether the analgesic drug reduces the time spent in the abnormal behaviours induced by the husbandry procedure. When the husbandry procedures under comparison differ in the type of pain they induce, the methodology of summing pain-related behaviours into a single class becomes somewhat less robust. Summing all the low-frequency behaviours indicative of pain into a single class assumes each type of behaviour occurs for about the same duration (i.e., the chance of observing each behaviour at an observation time point is equivalent) and that the importance of each behaviour as a sign of pain is also equivalent. For instance, ring castration increases the behaviour of lateral lying which is rarely seen in response to other castration treatments. Lateral lying is an inactive behaviour that might be over- represented at observation time points in comparison with short term active behaviours such as standing stretched, although the later might be associated with a more severe experience of pain by the lamb. Thus, when the husbandry practices under comparison induce differing spectrums of pain-related behaviours, the simple calculus of adding the counts of each pain-related behaviour into one group may introduce a bias into the methodology [76]. Nonetheless, when this caveat is kept in mind, the methodology provides a useful starting point for comparing husbandry practices. The impact of procedures on non-pain-related behaviours, such as grazing, on wound responses and physiological responses in the animal broadens the basis for comparison of procedures.

Use of telemetry and geolocation using movement trackers was effective at assessing pain in sheep with induced osteoarthritis [77]. In addition, stride length and number of steps were good indicators of post-castration pain in cattle [78]. With the development of new technologies, including the use of cinematic analyses [79], this provides potential for new practical measures that can be assessed over long periods with minimal effort.

A number of scoring systems have been developed to quantify pain-related behaviours. Termed Visual Analogue Scores (VAS), the behaviour scored and scale used are usually tailored to the type of pain-related behaviour under study, such as lameness [80]. In more holistic terms, pain can modify the demeanour (comportment) and activity level of the animal. These attributes of the animal can be assessed by the method termed qualitative behavioural assessment (QBA) without a deliberate focus when scoring the animal on any particular pain-related behaviour. QBA repeated across time has confirmed an association with flock health and welfare in sheep [81]. In cattle, QBA using free-choice profiling showed differences between castration with or without pain relief [82]. Development of rapid scoring systems may be valid for assessing the impacts of pain relief in livestock using QBA and VAS.

Ear postures were clearly shown to indicate pain in lambs undergoing tail docking by rubber ring [83,84]. The current manual method of assessing ear postures is time consuming and impractical. Development of technologies that can automatically measure ear postures may lead to a practical measure, but this has not been done yet.

#### 3.3.4. Physical Measures

Pressure mat readers have demonstrated that castrated calves shifted their weight forward onto their forelimbs when compared to sham-treated animals [85]. This type of measure that assesses weight placement on limbs is potentially a practical measure, is non-invasive and can be assessed over extended time periods [86,87].

#### 3.3.5. Cognitive Measures

Animals can form an association between an affective experience and a physical location in their environment. Through this process the environment can be described as becoming valenced [88]. This association can be assessed by conditioned place preference (CPP) testing. Individuals are conditioned to two locations that in an experimental setting may be distinguished by specific visual cues such as patterns or colours. One location is paired with a positive state such as a non-addictive drug treatment and the other location is paired with a neutral control treatment. Following a painful procedure such as ring castration, the time spent in each location provides insight into the animals’ experiences of pain in the two locations. A study on disbudding showed that calves were willing to trade-off the short-term aversiveness of local anaesthetic for pain relief, and that the pain of disbudding lasted for 3 weeks [89]. While CPP is not a practical method to apply on-farm, it is a useful experimental method for understanding the duration of pain and for measuring the effectiveness of pain relief treatments.

Judgement bias provides a measure of affective states and has been applied to assess the impact of hot iron disbudding on dairy calves. The study found that disbudding induced a pessimistic bias indicating that calves were in a negative affective state [90]. However, inconsistent findings were reported in sheep with the known stress of shearing inducing an optimistic bias when compared with control animals [91]. While this novel method provides interesting insights into animal feelings, it is impractical to apply, requires long training periods, and inconsistent findings have been reported in sheep, limiting its use in livestock.

A novel measure was investigated in a study in humans through assessing auditory sensory memory in relation to pain. The study found that mismatch negativity amplitude was reduced during pain exposure, indicating that chronic and acute pain can interfere with automatic change detection processes in the brain [92]. This approach has not yet been investigated in animals. However, there is potential to do so in future studies.

#### 3.3.6. Neurology

The acute pain of tail docking of lambs was identified using IRT and nociception assessment (indicators of inflammation), while histology was used to assess the chronic aspects of pain by assessing traumatic neuroma formation [93]. Using histology to inform on pain states is insightful as it provides a measure of the impacts of procedures on the more chronic aspects of pain. Histological measures were able to show that the pain associated with tail docking can last for 90 days.

Disbudded calves were assessed for morphological measures of neuropathology: diameter of axons in cornual and infraorbital nerves, glial fibrillary acidic proteins and activating transcription factor 3 [94]. No differences were evident, indicating that these may not be sensitive enough measures for assessment of pain in livestock.

#### 3.3.7. Vocalisation

Pigs that are castrated squeal louder and for longer than sham animals [95]. Vocalisation is a less useful measure in sheep as they are less likely to vocalise when in pain.

#### 3.3.8. Grimace Scale/Facial Expression

Preliminarily evidence suggests that in horses, the grimace (unnatural expression of the face) relates to pain, not emotional states [96]. In sheep, Guesgen et al. [97] found some evidence of changes in facial expression of lambs undergoing tail docking by rubber ring. However, this was confounded by the restraint procedure. In laboratory sheep undergoing surgery, the grimace scale was positively correlated with clinical findings [98]. Lu et al. [84] developed a facial action unit scale and found some evidence of its relation to pain. However, this was without any validation in relation to physiological measures. A recent review of facial expression for pain in mammals summarised that it is a promising measure, but feasibility testing and further refinement are needed to ensure reliability [99].

#### 3.3.9. Section Summary

A large range of measures can be used to provide insights into the pain response of animals, most of which focus on behavioural, physical or physiological changes. Insights into the subjective experience are more difficult to elicit. A major challenge in assessing the effect of analgesia is the large individual variation in response, and the fact that social and environmental context can modulate the responses. Biochemical markers may be of use in assessing sustained or chronic pain conditions, when behaviour has returned to near normal (an important survival strategy for prey species such as sheep). Novel measures are being developed, but many need further refinement to ensure repeatability and reliability. Major challenges remain in assessment of pain in a commercial setting. Further development of novel measures of pain that can be automated through using new (e.g., sensor) technologies to overcome the issues with practical application in the field is warranted.

The complex processes of induction and central processing, and consequent adverse outcomes suggest that measurement of pain will continue to rely on a combination of measures. A focus on developing a single unified measure is probably inappropriate because pain is multidimensional in terms of the experience and the consequences.

### 3.4. Sheep Husbandry Procedures

The bulk of the literature on the behavioural and physiological effects of the various husbandry procedures was published prior to 2000 and has previously been admirably reviewed [1,100,101,102,103,104,105]. More recent papers describing responses to castration and/or tail docking are mostly in the context of comparison against a test group receiving some form of analgesia, and merely confirm the previously reported findings. The following section provides a brief summary and update.

#### 3.4.1. Tail Docking and Castration

The majority of lambs produced in Australia are marked (castrated and tail docked) between the ages of 4 and 12 weeks. In a recent survey, 39% of sheep producers stated that they would be willing to use pain relief for castration and tail docking, and a further 20% said ‘maybe’ [106]. Of all producers (both sheep meat and wool producers) that castrate and tail dock (including those who do not mules), 31% of the producers were already using pain relief for mulesing. In a separate survey, specifically of woolgrowers, 84% of producers used pain relief for mulesing [107]. There is a variety of methods used for castration and tail docking, and although attempts have been made to compare differing methods, it is evident, particularly from behavioural assessment, that the responses to surgical and ischaemic methods differ entirely, such that they are exceedingly difficult to place in any sort of pain ranking.

##### Surgical Methods

Surgical castration is only carried out by 3% of Australian sheep producers, but hot knife tail docking remains popular, being used by 68–78% of specialist wool producers and 33% of specialist sheep meat producers [106,107]. Surgical castration or tail docking result in physiological changes including a marked elevation in plasma cortisol levels, acute phase protein responses and alterations in behaviour and posture [76,108,109,110,111,112,113,114]. Surgical tail docking also results in changes in gene expression within the nerve ganglia, suggesting that increased pain sensitivity may be present for 4 months or more post-procedure [3]. Few studies compare the different surgical approaches to tail docking (e.g., cautery/hot knife, cold knife), although in piglets, electroencephalographic studies have suggested that cautery tail docking is less acutely painful than docking using side cutters without cautery [13].

##### Rubber Ring (Ischaemic) Methods

Although surgical methods are available for castration and tail docking, use of constricting rubber rings is increasingly popular, particularly when lambs are non-mulesed [107]. A recent survey of Australian sheep producers indicated that 36% use rubber rings for tail docking [106], while a survey of woolgrowers indicated that 61% of producers that did not mules used rubber rings for tail docking, compared to 13% of producers that did mules [107], while for castration, both surveys indicated that 95–97% of Australian producers used rubber rings, whether they mules or not. The constricting rubber ring leads to ischaemic necrosis of the tissue which is subsequently sloughed, meaning that the procedure is bloodless. However, the process is acutely painful, leading to physiological changes and display of acute pain behaviours such as rolling and writhing on the ground, abnormal standing postures, and kicking or foot stamping [22,74,76,109,113,115,116,117,118,119,120,121,122]. Acute behavioural responses to applying rings usually resolve within one hour [76,123] whereas behavioural responses to knife treatments can last more than 4 h [76]. Nevertheless, reduced frequencies of lying and play behaviour and increased frequencies of abnormal postures in the three days post-procedure have been reported in lambs undergoing ischaemic castration [72], while alterations in physical activity and gait for up to 10 days post-procedure have been reported in calves undergoing ring castration [124], and alterations in tail posture and movements were observed 6 days post-rubber ring application to adult dairy cows undergoing ischaemic amputation of the tail [125]. There appears to be very little recent data available on the tissue physiology, particularly at the molecular level, of ischaemic pain associated with rubber ring application, other than some data demonstrating afferent nerve conductance for 90 min or more post-ring application [126]. The data that do exist on ischaemic pain are based on the situation in humans, where blood supply is restricted, e.g., as a result of cardiac insufficiency or thrombosis, in which the therapy is aimed at mitigating the pain until the restriction can be corrected, as opposed to the permanent severe constriction imposed by the rubber ring used for castration and tail docking. A deep understanding of the tissue physiology associated with rubber ring application would assist in selecting the most appropriate analgesic agent(s) for further investigation. Necrosis of the tissue distal to the constriction ultimately results in termination of nerve impulses from that tissue. However, there is also a tissue response at the junction between the healthy proximal tissue and the healing/sloughing stump. In lambs, lesions develop post-ring application, increasing in size, swelling, inflammation and visible signs of infection over 2–4 weeks post-ring application, and subsequently resolve [4,5,127,128], the severity and time-course of which are greater in older lambs. In lambs, there is an increase in pain-related activity in weeks 2–4 post-ring application, associated with the peak severity of these lesions [4,5]. Similarly, unhealed lesions, physiological and behavioural changes have been observed some 2–4 weeks post-band or -ring castration in cattle [6,7,8,111].

The clamp (burdizzo) method of tail docking or castration can also be considered an ischaemic method: application of the clamp across the scrotal neck or on the tail results in constriction of the nerves and blood vessels supplying the distal tissues, and possibly partial severing of the deeper structures, e.g., the spermatic cords. Subsequent removal of the clamp allows re-perfusion of the overlying skin, which may account for the observations that post-procedural pain-related behaviours are less marked with burdizzo castration than rubber ring [129]. Nevertheless, burdizzo methods also result in significantly elevated plasma cortisol levels on the day of the procedure [114,129,130].

##### Combined Methods

Some producers use a combination of methods, most commonly application of a rubber ring followed by surgical removal of the tail. A combination of rubber ring application followed by burdizzo clamping for both castration and tail docking has been evaluated in lambs aged from less than one week up to 6 weeks, with reductions in cortisol response and pain-related behaviours as compared with rubber ring alone [113,128,129]. Rhodes et al. [130] used burdizzo clamping followed by surgical removal of tails from 2–3 week-old lambs, the clamp being held on for 3 min following tail removal, and recorded a significant increase in cortisol as compared to handled controls. There were no other treatment groups in that study, so the authors could draw no conclusion other than burdizzo tail docking elicits a cortisol response.

#### 3.4.2. Mulesing and Its Alternatives

Based on surveys carried out in the period 2017–2018, mulesing is carried out by approximately 70% of Australian woolgrowers, 84% of whom use pain relief, and the vast majority of lambs are mulesed at the time of marking [107]. Across the broader Australian sheep industry, 21% of producers strip tails at the time of hot knife tail docking instead of mulesing [106]. Surgical mulesing, in the absence of analgesia, elicits profound physiological and behavioural responses, including marked reductions in activity and feeding, particularly on the day of the procedure, but aberrations in feeding behaviour, gait and growth for up to three weeks post-procedure have been reported [131,132]. Provision of pain relief, particularly use of a multi-modal approach, ameliorates these responses [60,133,134,135,136,137,138,139].

A variety of non-surgical approaches to breech modification have been evaluated, including application of occlusive plastic clips (an ischaemic approach), injection of necrotising agents and application of liquid nitrogen. Laser depilation as a means to increase the bare skin area has also been attempted, with limited success [140]. Published literature indicates that these alternatives still induce physiological and behavioural responses indicative of pain or discomfort, of a comparable or lesser degree (according to the specific measure used) to surgical mulesing [131,132,141,142,143], but it must be acknowledged that the majority of the research published relates to pre-commercial development research and does not necessarily reflect the final use pattern of the method assessed. There are very few publications evaluating the effect of pain relief on these alternative approaches: carprofen can reduce the behavioural response, but not the physiological responses to cetrimide injection [142]; but meloxicam appeared to have limited impact on the behavioural or physiological responses to liquid nitrogen application [144].

Other than genetic selection or gene editing approaches, all forms of physical modification of breech conformation will be associated with some degree of discomfort or pain. The behavioural and physiological responses to intradermal cetrimide or ‘mulesing clips’ have been compared to those of surgical mulesing without pain relief, but as at September 2019 no published comparisons against surgical mulesing with pain relief have been published.

#### 3.4.3. Other Procedures and Conditions

A variety of other procedures and conditions result in some degree of discomfort or pain: ear tagging; laparoscopic artificial insemination (AI), injuries and illnesses (e.g., shearing cuts, lameness, mastitis). In piglets, ear tagging and ear notching resulted in head shaking and scratching at the ear over a 3 h period post-procedure, and elevations in plasma cortisol and lactate levels [145]. Laparoscopy does induce a cortisol response in sheep, leads to expression of pain-related behaviours and postures, and is considered a painful procedure in humans [146,147,148]. Laparoscopic AI is carried out in sheep because transcervical AI is challenging to perform due to their anatomy. However, De Rossi et al. (2009) used subarachnoid ketamine; or misoprostol (a prostaglandin analogue) placed close to the cervix and intravenous oxytocin; or a combination of the two, prior to transcervical AI. All treatments provided sufficient cervical dilation to allow insertion of the insemination rod. Inclusion of ketamine eliminated pain/discomfort related behaviours and resulted in ataxia and sternal recumbency for 18–34 min [149]. No publications in the period 2000–2019 were found relating to pain associated with injury or illness in sheep.

#### 3.4.4. Section Summary

It is evident that the physiological and behavioural consequences of husbandry procedures, particularly in the case of the rubber ring methods of castration and tail docking, mulesing and alternatives to mulesing continue beyond the initial 24–48 h post-procedure, although demonstration of statistically significant differences between treatment groups is challenged by the large individual variation in responses.

### 3.5. Analgesic Agents

The majority of analgesic agents are pharmaceutical substances aimed at:Altering transduction, transmission and central modulation of nociception, either by direct disruption of nerve cell membranes, inhibition of the propagation of action potentials along nerve fibres, or through inhibition of one or more of the receptor types involved; orThrough inhibition of the production of inflammatory mediators such as prostaglandins or leukotrienes; orA combination of the above, many agents interacting with a variety of receptors.

Although a detailed discussion of pain physiology is beyond the scope of this document, it is worth introducing the main receptor types on sensory nerves. The transient receptor potential (TRP) ion channels are a group of chemo-, or thermo-sensitive channels including the transient receptor potential vanilloid (TRPV) channels, of which inhibition of TRPV1 has been shown to provide analgesic effects [2,150,151,152,153,154,155,156,157,158,159]. Acid-sensing Ion Channels (ASICs) detect changing extracellular pH, and are considered to be important receptors in ischaemic conditions of muscle, inflammation and tissue damage [58,160,161,162,163,164,165,166,167,168,169]. Voltage-gated sodium channels (Nav) open when the nerve cell is sufficiently stimulated to trigger an action potential and are vital in the transmission of the impulse along the nerve. There is some interest in blockade or inhibition of Nav channels as an analgesic approach [170].

It is important to note that for any agent:The rate of absorption differs by route of administration, formulation, agent and species;The rate of clearance of the agent from the body differs by agent and species (Table 4), which is related to the manner in which the different species metabolize the agent, and that in turn will influence the metabolites and potential residues present in different species;The minimum effective concentration of the agent in plasma can differ between species;Furthermore, there is large inter-individual variation within species, and age, physiological status and the presence or absence of an injury or insult can in turn affect the pharmacokinetic (time course of drug absorption, distribution, metabolism, and excretion) and pharmacodynamic (biochemical, physiological and toxicological, including dose) properties of an agent.

Therefore, extrapolation of dose rates and regimens from one species to another is not recommended, and the underpinning understanding of pharmacokinetics and pharmacodynamics (including absorption rates, clearance rates, metabolism and minimum effective concentration) of an agent in a particular species is vital to ensure effective analgesia is achieved. It is evident from Table 4 that much of this information is missing for livestock. Although the scope of this review excludes articles published prior to 2000, the initial search was broader, and there was no clear indication of this missing information having been published previously.

#### 3.5.1. General Anaesthesia

Although general anaesthesia is excluded from the scope of the current review, it is worth acknowledging the recent European interest in inhalational anaesthesia (e.g., using Isoflurane) for piglets undergoing tail docking. A restraint unit which clasps the piglet in dorsal recumbency with its head inserted into a chamber for delivery of the inhalational anaesthetic has been designed and is commercially available under the brand ‘Porc-Anest’^®^ (http://www.promatec.ch/48/porcanest, accessed on 15 March 2021). At this stage, it is unclear whether a similar unit could be designed for lambs; and if so, what volumes of inhalational anaesthetic would be needed for lambs, and the consequent economic, operator and environmental safety attributes of such a system are yet to be elicited.

#### 3.5.2. Local Anaesthetics

A significant body of research into the pharmacokinetics and efficacy of local anaesthetics in sheep was carried out prior to 2000 [74,116,123,214,215,216,217,218,219,220,221,222,223,224,225], and this interest has continued into the period between 2000 and 2019. Recent research includes the use of local anaesthetics as a component in a multi-modal approach to analgesia, aiming to optimise particular anaesthetic protocols. As sheep are often used as a model for human medical research, a number of the studies refining anaesthetic protocols have been carried out in sheep. Five local anaesthetics appear in current literature: Lignocaine (also known as lidocaine), bupivacaine, levobupivacaine, procaine and ropivacaine. Published research into the use of articaine, mepivacaine or proxymetacaine (proparacaine) in livestock was not found in the period 2000–2019.

Lignocaine is a rapid-onset (within 15 min) local anaesthetic which gives around 60 min of anaesthesia (dose and route of administration dependant) in sheep [226,227]. In cattle, caudal epidural administration of 0.2 mg/kg lignocaine leads to onset of tail paralysis within 2 min, analgesic effects beginning within 5–8 min and lasting for 25–130 min [228,229,230]. However, evidence of efficacy in livestock is patchy. Caudal epidural administration of lignocaine has been used as an anaesthetic approach for cattle undergoing surgical castration using an emasculator or burdizzo (clamp) castration, with no beneficial effects on cortisol response, behaviours during the procedures, gait parameters or behaviours on the subsequent two days [78,231]. Similarly, caudal epidural lignocaine did not mitigate the behavioural response to rubber ring tail docking in adult dairy cows on days 0, 2 and 6 post-procedure [125]. Local administration of 240 mg of lignocaine into the spermatic cords and scrotal neck prior to knife castration in calves did not appear to reduce the pain response as assessed using visual analogue scales (pain scores) [232]. Lignocaine does appear to be of benefit when calves are castrated using rings or bands: 100 mg of lignocaine delivered into the testicles and scrotum has been shown to obliterate the cortisol response to ring castration, but not surgical castration with traction, use of an emasculator, or burdizzo (clamp) castration [233], while 200 mg delivered into the spermatic cords and scrotal neck did result in a reduced cortisol response and reduced active pain behaviours following burdizzo (clamp) castration, and near obliteration of the cortisol response and pain behaviours following ring castration [234]. In piglets, lignocaine similarly seems to be of limited benefit. Subcutaneous injection of 6 mg around the base of the tail reduced the likelihood of the piglet squealing or struggling during cautery tail docking, but had no impact on post-procedure behavioural responses [235], while 40 mg injected into the testes and scrotal area led to a reduction in squeal volume and a slight reduction in cortisol response in piglets undergoing surgical castration [95].

In sheep undergoing castration, lignocaine appears to provide more benefit than for cattle and pigs, although the picture is muddied by a range of animal age, dose rates, administration routes and reporting protocols being used in published studies. There was a tendency to reduce visual analogue pain scores and cortisol response to burdizzo (clamp) castration, and a tendency to reduce visual analogue pain scores with a significant reduction in peak cortisol and active pain-related behaviours following ring castration in lambs less than 7 days old receiving 4 mg/kg into the spermatic cords and scrotal neck [236]. A needleless injector was used to deliver 8 mg of lignocaine into the testes, scrotal neck and tail of lambs less than 7 days old prior to ring castration, resulting in a reduced incidence of foot stamping/kicking and tail wagging [237], while 24 mg resulted in a significant reduction in active pain-related behaviours and visual analogue pain score [238]. In 4-week-old lambs undergoing ring castration, 120 mg lignocaine injected into the testes and scrotal neck at the time of ring application resulted in reduced lying, activity and postural changes compared to ring application without lignocaine [239]. No further benefit was observed by administering the lignocaine 4 min prior to ring application. Epidural administration of lignocaine at 2.2 mg/kg has been shown to attenuate the pain response to surgical mulesing [138], while in 8–9 month old goats, epidural administration of 4 mg/kg lignocaine obliterated the behavioural response to ring castration, induced recumbency for 132 min and attenuated the cortisol response for 120 min post-castration [240]. Lignocaine is considered to be poorly absorbed through intact skin, but even so, rubber lings coated in a lignocaine gel provided a degree of attenuation of both the cortisol and the behavioural responses to ring castration in 4-week-old lambs [239].

Bupivacaine has a longer (by up to three times) duration of action than lignocaine when given subcutaneously [226], as a nerve block [241,242], or epidurally [243,244]. The onset of effect of bupivacaine is slower than that of lignocaine (10–45 min post-administration c.f. 5–15 min in sheep) [242,243]. Levobupivacaine, an enantiomer of racemic bupivacaine with reported less cardiotoxicity and neurotoxicity, has similar rate of onset and duration of action as bupivacaine [245]. Both bupivacaine and levobupivacaine are considered safe, causing mild transient changes in blood glucose levels, heart and respiratory rate when given epidurally [245,246]. Lambs undergoing ring castration displayed lower levels of abnormal postures in the hours from 2.5 to 9 post-procedure when bupivacaine was used as compared with lignocaine (local anaesthesia delivered into each spermatic cord and around the scrotal neck) [110].

In the period 2000–2019, a single paper investigating the efficacy of 15 mg procaine delivered to the spermatic cords by needleless injector found a significant reduction in active pain behaviours and abnormal lying postures in lambs under 7 days old undergoing ring castration [5], and a single paper describing the pharmacokinetics of ropivacaine when given intravenously or epidurally [247] were found.

Given the relatively short duration of action of local anaesthetics, there has been some interest in adding substances to enhance the effect. Adrenaline (epinephrine) has long been used as a vasoconstrictor and is incorporated into some local anaesthetic formulations in order to reduce the rate of clearance of the agent from the site of injection. Recent studies demonstrate this for lignocaine with adrenaline injected epidurally, resulting in prolonged analgesia [229,243], while adrenaline added to either ropivacaine or bupivacaine alters the clearance and distribution processes within the epidural and intrathecal spaces, which could lead to better analgesic outcomes [248]. A variety of other additives have been evaluated: dexamethasone added to lignocaine for caudal epidural injection gave a significantly greater duration of analgesia in cattle than lignocaine alone (150–160 min c.f. 100–130 min), while in rats, a nerve block was extended from 2 h (bupivacaine alone) to 21 h (bupivacaine microspheres + dexamethasone) [241]; midazolam used with lignocaine for an intravenous regional block in humans reduced the time to onset of sensory and motor blockade, and provided prolonged analgesia than lignocaine alone, with lower reported post-operative pain scores [249]; butorphanol added to lignocaine for a lumbosacral subarachnoid injection prolonged the analgesia and ataxic effects [250] as did morphine added to bupivacaine [251]; neostigmine added to a lignocaine-bupivacaine mix for a nerve block in humans reduced the post-operative pain scores as compared to the lignocaine-bupivacaine mix alone [252]. The safety of adding either ketamine [246] or neosaxitoxin [253] to bupivacaine in a sheep model has also been evaluated, but no efficacy evaluations were found.

Combining lignocaine and bupivacaine, in order to reap the benefits of both (rapid onset of analgesia with lignocaine and prolonged duration of action with bupivacaine), has also received some attention in published literature. In sheep undergoing stifle arthrotomy, an intra-articular injection of 40 mg lignocaine and10 mg bupivacaine provided a significant reduction in pain score at 3–7 h post-operatively compared to the saline control [254]. However, from this study, we cannot comment on the comparison with either lignocaine or bupivacaine alone, as these control groups were not included. When a combination of 20 mg lignocaine and 50 mg bupivacaine was injected subcutaneously in sheep, nociceptive pressure sensitivity was reduced for 120 min [226]. This was superior to 40 mg lignocaine alone (60 min), but not to 100 mg bupivacaine alone (180 min). The authors did not thoroughly assess latency to onset of analgesia. The topical combination agent Tri-Solfen^®^ includes lignocaine, bupivacaine, adrenaline and cetrimide. It was initially registered for treatment of sheep undergoing surgical mulesing, and its pattern of use has been extended to include other painful procedures and conditions. In sheep undergoing surgical mulesing, it can provide significant reductions in pain-related postures (particularly hunched standing) in the first 2–4 h post-procedure, and attenuation of the cortisol response [60,133,134,139]. Longer term effects on pain-related behaviours are (unsurprisingly based on the expected duration of action of the local anaesthetics) limited [255], but a sustained effect on the development of mulesing wound hyperalgesia is a feature of Tri-Solfen use [60,136]. In surgical castration, when Tri-Solfen is sprayed onto the spermatic cords and cut edges of the scrotum, calves display reduced pain-related behaviour and the development of hyperalgesia of the wound site is similarly attenuated [59], although significant effects on the cortisol response were not evident [137]. In lambs undergoing surgical castration and cold-knife tail docking, Tri-Solfen reduced visual analogue pain scores, and prevented development of wound hyperalgesia [59], whereas in lambs undergoing surgical castration and hot knife tail docking, application of Tri-Solfen to the castration wound attenuated the cortisol response, but no benefit was observed associated with application of Tri-Solfen to the cauterized tail wound [256]. It is possible that the cauterized crust prevented absorption of the actives into the tissue on the tail.

##### Subsection Summary

Although local anaesthesia does provide amelioration of the acute pain response to painful husbandry procedures, the duration of effect is short lived. Combinations such as lignocaine with bupivacaine; addition of adrenaline, dexamethasone or Nav (voltage-gated sodium channel) blockers such as neosaxitoxin can somewhat prolong the duration of action of local anaesthetics, but it appears that durations of greater than 3–4 h are not currently achievable, with the exception of agent-impregnated microsphere technologies. However, it must be acknowledged that use of local anaesthetics prevents the development of hyperalgesia (wound sensitisation), such that wound sensitivity in animals treated with local anaesthetics is measurably reduced as compared to animals that did not receive local anaesthesia, for at least 24 h post-procedure.

#### 3.5.3. Non-Steroidal Anti-Inflammatory Agents (NSAIDs)

NSAIDs are anti-inflammatory and analgesic agents, that can also be anti-endotoxaemic, anti-pyrexic and anti-neoplastic. There are a number of mechanisms whereby their effects are produced, and these are not fully understood. The primary mechanism is through inhibiting the production of inflammatory mediators, such as prostaglandins and leukotrienes. Inhibition of prostaglandin production occurs as a result of the NSAID blocking the cyclooxygenase (COX) enzyme. To date, three isoforms of the COX enzyme have been identified. It was initially believed that the COX-1 enzyme function was part of normal biological homeostasis, and that COX-2 was associated merely with tissue pathology and inflammation. The importance of COX isoform selectivity relates to risk of adverse reactions, particularly associated with COX-1-mediated gastro-intestinal upsets, perforation and associated peritonitis, that have been reported in monogastrics [257,258,259,260,261,262] and it was assumed that a COX-2 selective agent would be free of such adverse effects. However, adverse cardiovascular, intestinal and other effects have been reported in humans or dogs associated with some COX-2 selective NSAIDs [259,262,263,264,265,266,267,268,269,270,271,272,273,274,275,276,277,278,279,280,281,282,283,284,285,286]. It is important to note here that COX-selectivity is a continuum, so an agent that is described as ‘COX-2 selective’ still has effects on COX-1, but the effects on COX-2 are evident at lower dose rates than the effects on COX-1 [275,276]. Furthermore, the COX-3 isoform has been identified, and its function and role is as yet unknown, although there is some suggestion that it might have a role in central modulation of inflammation [260,287,288,289,290,291,292,293,294,295,296,297,298,299,300,301,302]. Thus, the full picture regarding the physiology of cyclooxygenase inhibition and pain management remains unclear. A further challenge in investigating COX-selectivity is that data generated using in vitro and ex vivo protocols may not reflect the performance of the agent in vivo. For example, Eltenac was shown to be non-COX selective in ex vivo protocols using equine blood but was COX-2 selective in in vitro protocols [303]. To date, there seems to be little investigation into the expression of the various isoforms of the COX enzyme in sheep, and their function in modulation of the pain response to painful husbandry procedures.

Other mechanisms of action of NSAIDs include inhibition of the lipoxygenase (LOX) enzyme, inhibition of kinases (ERK), and inhibition of acid-sensing ion channels [58,160,163,164,165,167,168,304,305,306,307,308,309,310]. There is also a suggestion that NSAIDs may physically interfere with cell signalling through attachment to the lipid membrane of cells, and also that NSAIDs can interact with the endogenous opioid and endocannabinoid systems [288,311].

To date, two NSAID preparations have been registered for sheep in Australia, both containing meloxicam.

##### Salicylates (e.g., Aspirin)

Aspirin can be considered the ‘prototype’ NSAID, forming the basis from which all the other COX inhibitors have developed. It is non-selective, inhibiting both COX-1 and COX-2. The salicylates have received some recent interest as low-cost alternatives to the modern NSAIDs. In dogs, inhibition of both COX-1 and COX-2 isozymes by aspirin has been reported [312], although inhibition of blood clotting may contraindicate its use following surgical interventions [313]. Both intravenous and oral acetylsalicylic acid or sodium salicylate at 50 mg/kg have been used in a lameness model in calves and in calves undergoing surgical castration, but there was no significant attenuation of the cortisol response, lameness parameters or EEG responses to noxious stimuli [205,206,207]. Those authors also carried out pharmacokinetic evaluations, reported a very rapid clearance of the agent from plasma and suggested that neither the dose rate nor frequency of administration used were sufficient to achieve effective pain relief. In sheep, a pharmacokinetic study at doses between 10 and 200 mg/kg suggested that doses of 100–200 mg/kg may be sufficient to produce effective analgesia [204]. However, the minimum effective plasma concentration in sheep is not known.

##### Enolic Acid Derivatives (e.g., Meloxicam and Piroxicam)

Meloxicam has received substantial interest in recent years, with 71 separate articles considering its efficacy, pharmacokinetics, pharmacodynamics and toxicity in a range of species included in the current review [82,86,87,95,133,134,175,176,188,189,190,191,192,193,194,195,196,197,198,199,200,201,202,203,235,255,257,258,259,306,307,312,313,314,315,316,317,318,319,320,321,322,323,324,325,326,327,328,329,330,331,332,333,334,335,336,337,338,339,340,341,342,343,344,345,346,347,348,349,350,351,352]. Meloxicam is considered to be COX-2 selective [306,307,312], and has minimal effects on platelet function, making it suitable for post-operative use [313]. Despite its COX-1-sparing reputation, it is not excluded from harmful side-effects such as gastric perforation [259]. To date, no information on the potential for NSAID-associated intestinal perforation in ruminants has been found.

The elimination half-life of meloxicam is long (Table 4), making it a practical agent for use in livestock, and dose rates of 1 mg/kg and above have shown good analgesic efficacy in a lameness model in sheep and in lambs undergoing surgical mulesing or knife castration and tail docking [86,133,134,188,348]. Some studies have attempted to use dose rates of 0.5 mg/kg in sheep, with limited analgesic effect [86,344], although pharmacokinetic data suggest that good plasma concentrations can be achieved at a dose rate of 0.5 mg/kg [190,191]. The apparent contradiction between the pharmacokinetic data and efficacy studies is probably related to the minimum effective concentration of meloxicam in sheep as compared to other species: a parameter that has not been identified in sheep.

No studies relating to the use of piroxicam in livestock were identified between 2000 and 2019.

##### Benzones (e.g., Phenylbutazone)

A single publication relating to benzones were found in the period 2000–2019. Phenylbutzone, delivered intravenously at 8 mg/kg in sheep had no effect on mechanical nociceptive threshold [353].

##### Pyrazolones (e.g., Metamizole)

No publications evaluating the analgesic effect of metamizole (dipyrone) in livestock were found in the period 2000–2019. A publication relating to the pharmacokinetics of metamizole in sheep, found that elimination half-life of the active metabolite was 3.01 h when metamizole was given intravenously, and 1.45 h when given intramuscularly [354]. Those authors considered that there were no clinically relevant differences in the metabolite pharmacokinetics after intramuscular or intravenous administration of metamizole. In goats, the active metabolite reached peak plasma concentrations 2.16 h after intramuscular injection of metamizole, and had an elimination half-life of 7.37 h, compared to 3.91 h for intravenous injection, leading the authors to recommend intramuscular injection as the better administration route in goats [355].

##### Propionic Acid Derivatives (e.g., Carprofen, Flurbiprofen, Ibuprofen, Ketoprofen, Naproxen and Vedaprofen)

Carprofen and ketoprofen exist as registered products for cattle in Australia, and as such have received some research interest. Carprofen, in particular is reported to remain at high levels in plasma for a prolonged period [356], although pharmacokinetic data for carprofen in sheep were not found in the published literature between 2000 and 2019. In sheep, carprofen at 4 mg/kg given either subcutaneously or intramuscularly 90 min prior to the procedure reduced the cortisol and behavioural response to surgical mulesing [138,139], but provided only modest analgesia to lambs undergoing surgical castration and hot knife tail docking [256]. A dose of 0.5 mg/kg administered subcutaneously was not effective at ameliorating the pain response to rubber ring castration and tail docking in lambs [357], and neither did a dose of 1.4 mg/kg administered intravenously to calves ameliorate the cortisol and inflammatory cytokine responses to either band or clamp (burdizzo) castration [69].

A number of studies have assessed the potential for 3 mg/kg ketoprofen given intramuscularly to cattle to mitigate the pain response associated with band or surgical castration, with little success [6,7,358,359]. The pharmacokinetic data for ketoprofen in cattle suggest that it is rapidly cleared from plasma, with an elimination half-life of less than one hour [181], which could account for its apparent failure to mitigate the pain responses to castration. The elimination half-life of ketoprofen in sheep is similar or slightly longer than in cattle, and suppression of PGE2 and TXB2 production for up to 12 h following a single intravenous injection of 1.5 or 3 mg/kg has been reported [179]. Nevertheless, a single efficacy study in sheep failed to demonstrate mitigation of the pain response associated with surgical mulesing when 4 mg/kg ketoprofen was administered subcutaneously 90 min prior to the procedure [138].

Ibuprofen has been studied in sheep, using sheep as a model for experimental embolization of the uterine artery (a treatment for uterine fibroids in humans, that has been associated with moderate to severe post-procedural pain) using ibuprofen-impregnated microbeads. Tissue samples indicated that the ibuprofen reduced the inflammatory response to embolization, and did indeed migrate into the tissues surrounding the embolus [178,360,361].

##### Acetic Acid Derivatives (e.g., Diclofenac, Eltenac, Etodolac, Indomethacin and Ketorolac)

In sheep, diclofenac has an elimination half-life of 2.12 to 2.84 h [174], and ketorolac less, 14.85 to 17.85 min [186] suggesting that repeated dosing within a 24 h period would be required to provide effective analgesia using either agent. Eltenac and etodolac have not been studied in sheep, although research in dogs and horses suggest that they may be COX-2 selective in these species and significant suppression of PGE2 and TXB2 production can be achieved [303,362].

##### Fenamic Acid Derivatives (e.g., Meclofenamic Acid and Tolfenamic Acid)

Pharmacokinetic data for tolfenamic acid in sheep were not found, although the elimination half-life in goats (2.29 h) was found to be significantly shorter than in cattle (8.22 h) [212]. A dose of 2 mg/kg, administered intramuscularly 45 min prior to, or at the time of, surgical mulesing had no effect on pain-related behaviours, cortisol or β-endorphin response [344], although when administered intravenously, 2 mg/kg tolfenamic acid can reduce the mechanical nociceptive threshold in sheep [353]. Mefenamic acid has not been studied in sheep, and a single article on intraperitoneal administration in laboratory mice indicated that it had suppressive effects on both the cell-mediated and humoral immune systems [187].

##### Coxibs (e.g., Deracoxib, Firocoxib, Mavacoxib and Robenacoxib)

The coxibs are used widely in humans and dogs, with good effect [313,326,346,362,363]. They are markedly COX-1 sparing, but still are not exempt from severe adverse side-effects [262,284]. Oral cimicoxib has been evaluated in sheep, showing slow and very variable absorption rates, which may be associated with the relative insolubility of the agent [364].

##### Acetaminophen (Paracetamol)

Paracetamol has been in use for over 100 years, but its mechanism of action is still unclear. The controversy lies in particular with regard to whether it acts centrally or peripherally, or both, as evidence has been presented suggesting that it not only inhibits the COX enzyme, but also interacts with the endogenous opioid and endocannabinoid systems [289,311,365,366], without leading to the classic patterns of addiction associated with opioids and psychoactive cannabinoids. To date, there appears to be little data published on the pharmacokinetics, toxicity or efficacy of paracetamol in sheep, although two papers were found describing pharmacokinetics in calves, camels and goats [171,367].

##### Pyridinemonocarboxylic Acids (e.g., Flunixin)

A single pharmacokinetic study for in-feed flunixin administration, with a reported elimination half-life of 7.94 h [177], was found in the period 2000–2019. In contrast, there were a number of papers investigating the efficacy of flunixin in cattle and sheep. In calves an intravenous injection of 1 mg/kg reduced lameness parameters in a controlled lameness model study [368], while 3.33 mg/kg in a pour-on formulation significantly reduced the cortisol response to surgical castration [85]. Local administration of flunixin at the site of band castration did not alleviate the pain response in cattle [369], but was effective in attenuating the cortisol response and reducing pain-related behaviours and postures in ring-castrated lambs [345]. In surgically mulesed lambs, 2.5 mg/kg flunixin, given 90 min prior to the procedure either subcutaneously or intramuscularly, reduced the overall cortisol response, reduced hunched standing behaviour and increased time spent lying and walking normally [138,139], while in lambs undergoing surgical castration and hot knife tail docking, 2 mg/kg flunixin intramuscular or 4 mg/kg in feed both resulted in a reduction in pain avoidance behaviours in the first hour post-procedure and a reduction in abnormal postures on the day of the procedure [370].

##### Hydroxamic Acids (e.g., Tepoxalin)

Tepoxalin is unusual in that it inhibits the LOX enzyme as well as COX-1 and COX-2, and has been used in dogs to alleviate osteoarthritis, uveitis, dermatitis and post-operative pain [306,307,331,346,371]. Although the pharmacokinetics and pharmacodynamics of tepoxalin in horses [209,303], rabbits [210], chickens [208,211,372], dogs [373] and cats [307] have been investigated, there appears to have been no evaluations of pharmacokinetics, efficacy or toxicity in sheep or cattle.

##### Subsection Summary

Based on the pharmacokinetic data available, it would appear that, of the NSAIDs, meloxicam is the most practical agent of choice for use in the sheep industry, having a long elimination half-life of between 10.5 and 15 h. Flunixin follows, with an elimination half-life of 7.95 h when provided in feed, and carprofen may also provide sustained analgesia, although the pharmacokinetic data in sheep are not available. The minimum effective plasma concentration for any of these agents has not been reported, although effective pain relief has been demonstrated for meloxicam at 1.0 mg/kg, and for flunixin at 2.0 mg/kg or more.

#### 3.5.4. Sedative Agents

A number of agents that are traditionally considered ‘sedative’ or used as part of an anaesthetic protocol have been shown to have some analgesic effect. The biggest challenge to using these agents in the field is their psychoactive properties, and thus their potential for use as recreational agents for humans. Hence, regulatory controls are likely to limit the use of these agents in the field.

##### α-Agonists

The α-agonists are commonly used as sedative agents in livestock, but analgesic effects have also been demonstrated, particularly at below-sedative doses. At sedative doses (0.4 mg/kg xylazine in sheep), some of the findings must be interpreted with care: if a reduction in nociceptive response is recorded coincidental with moderate to deep sedation, but not when no or mild sedation is observed [325,374], is this truly evidence of analgesia, or merely evidence that the animal is unable to, or is very slow to, respond to the stimulus? However, at very low doses, reductions in nociceptive threshold (pinprick or electrical stimulus) have been reported [37,227,228,375,376,377,378,379] when xylazine (0.05 mg/kg), clonidine (0.002–0.005 mg/kg) or detomidine (0.01 mg/kg) have been administered intravenously, intramuscularly, intraperitoneally or epidurally. Similarly, 0.003 mg/kg/h medetomidine administered by continuous intraperitoneal infusion has been shown to reduce post-operative pain without inducing sedation in sheep undergoing experimental surgical procedures [380].

Although low-dose α-agonists have shown good potential as analgesic agents under laboratory conditions, and can provide good post-operative analgesia, demonstration of analgesic effect in a field situation is a bit more challenging. In cattle undergoing surgical castration (traction), 0.05 mg/kg xylazine given intravenously 30 min prior to the procedure had an elimination half-life of 12.9 min and significantly reduced the cortisol response [318], but had no effect on the behavioural response to the procedure. Adding 0.1 mg/kg ketamine to the sedative injection led to a significant reduction in cortisol response and in behavioural response to the procedure. The elimination half-lives of the agents were 11.2 min (xylazine) and 10.6 min (ketamine) when given in combination [381]. In sheep, although reduced behavioural response at the time of surgical mulesing was observed, overall analgesia for surgical mulesing using epidural xylazine (0.15 mg/kg) alone was not considered satisfactory [138], while in goat kids 0.1 mg/kg dexmedetomidine suppressed pain-related behaviours during disbudding and suppressed the cortisol response [342].

Clonidine is of interest because it appears to have a different effect on damaged central nervous tissues than on undamaged tissues, enhancing depolarization-induced acetylcholine release in neuropathic but not normal spinal cord tissue, which suggests that it may have enhanced analgesic effects in treating neuropathic pain [382]. The duration of analgesia provided by epidural clonidine (118–311 min) can be significantly longer than xylazine (88 min) or lignocaine (65 min) [230,383].

The combination of an α-agonist with ketamine is a common sedative protocol in livestock, but should be used with some care, particularly when managing pregnant females: the agents readily cross the placenta and will affect the foetus as well as the dam [384].

##### Dissociative Agents

Ketamine has been used as a component in a multi-modal approach to anaesthesia/analgesia. Local infiltration of ketamine (0.5 or 1 mg/kg) has been shown to reduce the post-operative pain associated with tonsillectomy in children [385] and pain associated with surgical incisions [386], while addition of 1.5 mg/kg ketamine as a lumbar epidural adjunct to general anaesthesia reduced the need for post-operative opioid analgesia and lameness score in sheep undergoing hindlimb orthopaedic surgery [387]. In sheep, the analgesic efficacy and safety of ketamine delivered intrathecally or epidurally has been studied between 2000 and 2019. Caudal epidural 2.5 mg/kg ketamine resulted in a transient rise in blood glucose, but no changes to other haematological or serum biochemistry parameters [246]. Intrathecal infusion of 40–100 µM ketamine had no major effect on nociceptive response threshold for up to 3 h post-infusion, but reduced hypersensitivity to NMDA infusion [378]. Lumbar epidural injection of 1.5 mg/kg ketamine resulted in complete analgesia within 5 min of injection, lasting for 30 min, and sheep could be transcervically inseminated without signs of pain or distress [149]. In that study, addition of oxytocin and misoprostol did not provide further benefit, but addition of 100 mg magnesium sulphate to an injection solution of 2.5 mg/kg ketamine prolonged the duration of lumbar epidural anaesthesia from 41 to 115 min [388].

##### Benzodiazepines

Johansen et al. [389] evaluated the safety of lumbar intrathecal midazolam in sheep. Doses of up to 15 mg/day were continuously infused and were well tolerated, with behaviour and neurological function remaining normal and no significant effects on clinical parameters including haematology and blood biochemistry. No other publications relating to the analgesic effects of benzodiazepines in livestock were found in the period 2000–2019.

##### Subsection Summary

Although there has been some interest in the use of sedative agents as part of a multi-modal approach to livestock analgesia, they are psychoactive agents and therefore at risk of being used for recreational abuse, so it is unlikely that they will enter mainstream use in livestock production.

#### 3.5.5. Opioids

Opioids are considered to act predominantly on the central nervous system, although opioid receptors can be expressed on peripheral nerves, particularly during inflammation. Activation of the peripheral opioid receptors reduces excitation of the nerve. The central opioid receptor system is involved in regulating respiration, appetite, body temperature, stress and pain. Receptors are classified as µ-receptors (of which µ1 activation leads to profound pain relief, while µ2 activation impacts on heart and respiratory function), κ-receptors (activation of which can lead to pain relief and mild sedation, but with less respiratory and cardiac impact than µ-receptor activation), and δ-receptors (activation of which can lead to pain relief). Opioids are classed as ‘agonist’, ‘agonist-antagonist’ (also known as ‘partial agonist’) and ‘antagonist’, depending on their mode of interaction with the opioid receptors. Opioid agonists include morphine; meperidine (pethidine); hydromorphone; oxymorphone; methadone; fentanyl; remifentanil; sufentanil; alfentanil; tramadol and tapentadol. Opioid agonist-antagonists include butorphanol and buprenorphine.

One of the challenges with centrally active agents is delivering sufficient concentrations to the central nervous system—although high plasma concentrations may be present, analgesic effect can be limited if insufficient crosses the blood–brain barrier. The slow rate of diffusion of the agents across the blood–brain barrier could account for the slow onset of analgesia relative to plasma concentrations of the agent [390,391,392]. Lalani et al. [393] identified that the bioavailability of tramadol to the brain of mice was only 75% when a tramadol solution was administered intranasally. However, when tramadol was prepared as a microemulsion or nanoemulsion, the in-brain bioavailability of intranasal tramadol was 300% and more. Careful formulation of pharmaceutical agents is vital to maximise their potential for efficacy.

Fentanyl can be administered transdermally using patches or a transdermal fentanyl solution (Recuvyra^®^, Elanco, IN, USA). Although the solution led to severe adverse effects in sheep [394], transdermal fentanyl patches have been shown to provide good sedation and analgesia for 72 h in sheep and goats undergoing spinal or orthopaedic surgery [395,396,397]. The sedation aspect would pose challenges in terms of moving animals post-procedure in a commercial situation. In pregnant animals, the use of fentanyl patches may be contraindicated by the fact that the agent is transferred across the placenta into the foetus [398]. De Souza et al. [399] investigated the pharmacokinetics of 2 mg/kg tramadol in goats following either oral or intravenous administration, and found that oral administration was likely to be ineffective in goats, while intravenous injection would need to be repeated every 6 h to maintain plasma levels equivalent to those found to provide effective pain relief in humans. Intravenous tramadol at 4 or 6 mg/kg was used in sheep undergoing spinal surgery with no adverse effects [400], but no assessment of analgesic effect was conducted, while in cattle, tramadol, administered either intravenously (4 mg/kg) or as a suppository (100 mg), did not reduce the behavioural responses to caustic paste disbudding.

Buprenorphine and butorphanol are widely used in companion animal medicine as part of multi-modal anaesthesia-analgesia strategies. Pharmacokinetic studies have shown that an intranasal application of buprenorphine resulted in high bioavailability [401], while a subcutaneous sustained release formulation resulted in slow onset, but long duration of plasma concentrations assumed to be therapeutic in sheep [402,403], although the authors acknowledge that the minimum analgesic threshold concentration of buprenorphine is as yet unconfirmed. In lambs undergoing surgical mulesing, 12 µg/kg buprenorphine reduced hunched standing and total abnormal behaviours or postures, and increased the time spent lying and walking normally [138]. In llamas, butorphanol, given intramuscularly at 0.1 mg/kg, can reduce nociceptive response threshold to pressure (in the absence of injury), but analgesic effect has not been validated using a pain model. In the same paper, the authors reported the pharmacokinetic parameters of 0.1 mg/kg butorphanol given intramuscularly or intravenously and concluded that the elimination half-life using the intravenous route was so short as to potentially limit the clinical usefulness of the agent [404].

##### Subsection Summary

The opioids are extremely effective analgesics, but the onset of action is delayed following administration due to the need for the agent to cross the blood–brain barrier. Although there has been some interest in the use of opioids for sheep undergoing invasive surgery, often in the context of biomedical research, it is unlikely that opioids will enter mainstream use in livestock production due to concerns over human dependence (addiction) and recreational use of such agents. Selective µ-opioid agonists may induce less dependence, but are still in the early stages of development [405].

#### 3.5.6. Gabapentinoids

Predominantly considered anti-epileptic drugs, gabapentinoids also have analgesic properties and are used to address neuropathic pain. Gabapentinoids block calcium channels, which appears to interrupt neurotransmission and excitability of nerve cells [406]. Although they can reduce post-injury hyperalgesia, and therefore may have a role in mitigating post-operative effects, they seem to have little effect on the acute pain response [407]. In livestock, gabapentinoids have been investigated as an adjunct to local anaesthesia in cattle undergoing scoop dehorning, with inconclusive results [175,176], and the pharmacokinetics of gabapentin co-administered with meloxicam have been studied in lactating dairy cows and calves [201,323]. No publications investigating the analgesic efficacy of gabapentinoids in sheep were found in the period 2000–2019.

### 3.6. The Future of Analgesia

From the early 1990s, research attention has focused on the specific molecular-level receptors and ion channels that are involved in pain signalling and transduction. Sodium ion channels, calcium channels, transient receptor potential channels and acid-sensing ion channels are being researched as possible targets for analgesia, while the role of the endocannabinoid system in pain modulation is also receiving much interest. The following section attempts to summarise the current state of understanding of some of these channels and systems, categorised as described in the literature. It is becoming increasingly clear that there is substantial overlap between these categories, with agents that affect one also affecting others. As our understanding of the physiology develops, the distinctions between the categories may become blurred and ultimately be consigned to history.

#### 3.6.1. Transient Receptor Potential Channels

There has been much interest recently in the role of transient receptor potential (TRP) channels in the pain and nociception pathways, and their potential as targets for novel analgesics. TRPs are a large group of cell membrane ion channels that are gated by a wide range of stimuli such as temperature, mechanical pressure or chemicals (ions to small molecules) including the pro-inflammatory agents such as prostaglandins, bradykinin, histamine and a variety of other trophic factors [158,159,408,409]. The TRPs are subdivided into subfamilies: vanilloid (TRPV); melastatin (TRPM); polycystin (TRPP); mucopilin (TRPML); canonical (TRPC) and ankyrin (TRPA), based on their structure. The contributions of the TRPs to pain sensation are not fully understood, but evidence is accumulating that blockade of TRPs, particularly the TRPVs, can reduce pain and hyperalgesia. TRPV1 has received the greatest research interest to date, but other TRPs are involved in pain physiology, so new approaches to analgesia, targeting these receptors, may develop. A detailed discussion of the current understanding of the TRPs in pain modulation is beyond the scope of this review, but a number of recent comprehensive reviews are available [158,159,409,410,411]. Key points are summarised here, and it must be acknowledged that the research cited is a mere fraction of the literature available.

#### 3.6.2. Vanilloid Receptor Antagonists

The vanilloid receptors are found in both the central and peripheral nervous system and are involved in transmission and modulation of pain. In nerve cells, calcium ions flowing through the activated TRPV1 channel lead to opening the voltage-gated sodium channels that precipitate generation of a nerve impulse (action potential). TRPV1 and TRPV3 have been associated with a variety of pain conditions including nerve injury [412,413] (including constriction-type), neuralgia and inflammatory conditions, in which case TRPV antagonists may be of benefit post-procedures such as ring castration and tail docking, in which there is constriction, nerve damage and subsequent inflammation as the tissue necroses and sloughs. TRPV1 is activated by reduced pH, so may have a role in the pain of ischaemia [414]. Antagonists of TRPV1 have been shown to reduce pain-related behaviour in experimental models of inflammation and osteoarthritis in laboratory rats [153,415,416,417,418,419]. However, in preclinical trials of the first potential analgesic candidates in humans it was found that blocking TRPV1 resulted in an undesirable increase in body temperature [420], and it appears that development of a commercial product has slowed, at least until an effective agent that lacks the undesirable side-effects can be found [150,421].

#### 3.6.3. Vanilloid Receptor Agonists

Counterintuitively, TRPV1 agonists can also be used to mitigate some forms of pain, through desensitising the channel or reducing the responsiveness of the sodium-channels that precipitate nerve depolarisation. Capsaicin (from hot chilli peppers) and resiniferatoxin (RTX, from one of the *Euphorbia* plant species) are pungent activators of TRPV1. In laboratory models, capsaicin and RTX have been demonstrated to reduce pain-related behaviours following incisional, thermal (burns) or inflammatory insults [422,423,424,425,426,427]. In humans, a dermal patch containing 8% capsaicin has provided prolonged (3 months or more) relief of post-herpetic neuralgia (residual pain after shingles) [428]. The capsaicin itself is irritant, so local anaesthetic gel may need to be applied to the skin prior to patch application. For this reason, research continues into alternative vanilloids that have reduced irritant properties. Some of these have been shown to reduce inflammatory hyperalgesia and pain in rats, with potential durations of effect of a week or more, particularly those compounds that can access the central nervous system [153,415,417,418,427,429]. Clinical trials in humans (reviewed by Remadevi and Szallasi, 2008 [430], and Wong and Gavva, 2009 [421]) indicated that the vanilloid Adlea^®^ (Anesiva Inc., San Frasisco, CA, USA) provided pain relief for up to 4 days post-surgery [431,432], while in patients suffering from arthritic pain, relief for up to 4 weeks has been reported.

A variety of other natural compounds also activate TRPV1: piperine (black pepper); zingerone (horseradish); eugenol (cloves); gingerols and shogaols (both from ginger); allicin (garlic, onion); camphor [433,434,435,436,437,438,439]. Both camphor and eugenols are commonly used in topical analgesic balms and liniments. Eugenol, the active ingredient of clove oil, also inhibits calcium and sodium channels in nerve cells, contributing to its analgesic effects [440,441]. It may be of use as a topical analgesic for ear tagging or shearing cuts, but this is yet to be assessed.

#### 3.6.4. Calcium-Channel Blockers

Voltage-gated calcium channels (VGCCs) are involved in many biological processes, including inflammation and pain. To date, research seems to have focused mainly on understanding the role of VGCCs in visceral pain, with VGCC blockers such as nifedipine or diltiazem delivered directly into the brain resulting in reductions in reticulo-ruminal disorders and general nociceptive behaviour, particularly tachycardia and hyperventilation associated with intestinal distension in sheep [442,443,444,445]. Eugenol, which is used as a topical analgesic in dental practice has also been shown to inhibit VGCCs, which may contribute to its efficacy [440]. The gabapentinoids also exert their effect through targeting a certain part of the VGCC.

#### 3.6.5. Sodium-Channel Blockers

The sodium channel blockers may provide greater durations of analgesia than current local anaesthetics, used either alone or in conjunction with a local anaesthetic. Amitriptyline and N-methyl-amitriptyline given as a sciatic nerve block or intrathecally have been shown to provide nociceptive blockade lasting for 7–11 h in rats [241,446] and 4–6 h in sheep [446], substantially longer than lignocaine or bupivacaine. QX-314, a quaternary derivative of lignocaine that acts on the sodium channels, prolonged the duration of sciatic nerve block analgesia in rats and mice for up to 9 h when combined with lignocaine [29,170]. Administered separately, lignocaine provided analgesia for an hour, and QX-314 was ineffective in producing analgesia [170]. Some agents have a combined effect on multiple channels or receptors: eugenol has effect on sodium channels, calcium channels and TRPV1, and it is likely that all contribute to its analgesic effect [435,440,441].

#### 3.6.6. Acid Sensing Ion Channels (ASIC)

ASICs play a role in nociception in both the peripheral and central nervous systems. ASIC gene expression is upregulated during inflammation [167,308,447], and increases in expression can be prevented using some NSAIDs, e.g., aspirin, diclofenac, and flurbiprofen [309]. Amiloride (a potassium-sparing diuretic) can block pain associated with acidic insult to the skin [448], while APETx2 (a peptide from sea anemones) also blocks ASICs, reducing pain-related behaviour and development of hyperalgesia in rat models of surgery or osteoarthritis [164,165,166]. The development of ASIC inhibitors is in its infancy, but the evidence is that ASICs are important components of the pain signalling pathway. ASIC inhibitors may prove to be highly useful in ischaemic conditions (e.g., ring castration and tail docking), in which tissue acidosis is likely to be a feature.

#### 3.6.7. Proteinase Activated Receptors (PAR)

Proteinase enzymes can act as signalling molecules by triggering G-protein coupled receptors. Activation of PAR1 prevents the development of hyperalgesia in laboratory models of inflammation, possibly through activation of the endogenous opioid system [28,449,450]. However, activation of PAR2 increases pain and inflammation [451,452], and the search is underway for effective PAR2 antagonists [453].

#### 3.6.8. Cholinergic Receptors and Acetylcholinesterase

The cholinergic system also seems to have a role to play in pain signalling and modulation, but it appears that research into potential analgesic agents is still at a very early stage. Muscarinic cholinergic agonists and acetylcholinesterase inhibitors such as neostigmine and bethanechol activate muscarinic receptors in the spinal cord, which can lead to a reduction in nociception that lasts for up to 2 h after intrathecal injection [454]. No information was found pertaining to administration routes other than epidural approaches. Spinal nicotinic cholinergic receptors are also involved in the pain pathway: intrathecal nicotine increased sensitivity to thermal and electrical nociceptive tests in rats, while a nicotinic receptor blocker reduced its effect [455].

#### 3.6.9. Cannabinoids

The endogenous cannabinoid (endocannabinoid) system involves the central and peripheral nervous systems, and the immune system, all of which play a part in pain signalling and modulation. Herbal cannabinoids have long been used as analgesics and recent developments in our understanding of the molecular processes involved in the endocannabinoid system have stimulated interest in this as a target for new analgesic approaches, particularly for chronic and neuropathic pain. A systematic review of published studies and clinical trials found that cannabinoids are not effective in mitigating acute pain (e.g., post-operative pain) [456], so their usefulness to the livestock industries may be limited.

#### 3.6.10. Other Potential Targets

A variety of other receptors, enzymes and cytokines that play a role in the modulation of pain and inflammation have been identified and are being researched as potential targets for novel analgesics. These include, but are not limited to, angiotensin II, glycine, glutamate, GABA and protein kinase [155,457,458,459,460,461,462,463,464,465,466,467,468,469,470,471,472,473,474]. A detailed discussion of this literature to date is beyond the scope of this review.

#### 3.6.11. Section Summary

The ion channels and molecular receptors recently identified show promise as targets for new analgesic agents. However, development of these agents is still predominantly in its infancy, and although research can be carried out using individual chemicals prepared in the laboratory, it will be a number of years before a formulation can be made available for the livestock industries. From initial formulation, the regulatory process requires safety, toxicity, tissue residue and efficacy studies prior to registration of the formulation.

Cannabinoids show potential in treatment of chronic intractable pain conditions, such as neuropathic/neurogenic pain. However, they do not appear to be effective analgesics for acute and post-operative painful conditions, so their relevance to the livestock industries is likely to be limited.

### 3.7. Multi-Modal Analgesia

Multi-modal analgesia is the practice of using more than one agent so that the benefits of each can be obtained. In human surgery, a patient may receive: pre-operative NSAID, pre-anaesthetic opioid and benzodiazepine; general anaesthesia induced using an injectable agent and maintained using a combination of inhalational agents (e.g., isoflurane plus nitrous oxide); intraoperative regional anaesthesia via an epidural spinal block or peripheral nerve block; intraoperative local anaesthesia, or application of another analgesic agent at the surgical site; post-operative analgesia using opioids and/or NSAIDs. The use of multiple analgesic approaches in this case both improves the overall post-operative pain levels for the patient and reduces the dose of each agent required, thus reducing the risk of toxicity [475,476,477,478]. Although this level of multi-modal analgesia is achievable and to be recommended for livestock undergoing surgery in a biomedical research context, it would be both practically and economically unachievable in routine livestock production.

Use of local anaesthesia with NSAIDs, for livestock undergoing routine husbandry procedures does provide greater amelioration of the pain response than use of a single agent alone [133,134,255] and should be recommended as current best practice. It may be of value to industry, supporting adoption, to continue to systematically evaluate multi-modal approaches to the various husbandry procedure methodologies (e.g., surgical or ischaemic) and combinations of procedures (e.g., mulesing with castration).

Much of the recent research into pain and analgesia focuses on the *treatment* of chronic pain conditions, e.g., neuropathic pain. In light of the finding that cautery tail docking, in piglets, can lead to sustained changes in the spinal tissues [3], there would be value in exploring the potential for these novel analgesics to *prevent* the development of these changes. If that can be achieved, such agents could provide a valuable addition to a multi-modal approach to management of pain associated with husbandry procedures.

### 3.8. Alternative Analgesic Modalities

Potential methods of providing analgesia to mammals that are not reliant on administration of pharmacological agents include: transcutaneous electrical nerve stimulation (TENS); electroacupuncture; interferential analgesia and other electrical modalities; photomodulation or electromagnetic analgesia; and cryoanalgesia.

#### 3.8.1. TENS

Transcutaneous electrical nerve stimulation machines deliver small electrical pulses to the body via electrodes placed on the skin, incorporating either flexible rubber pads or self-adhesive pads placed over a thin conductive gel. There are a variety of TENS-like devices providing variation in current amplitude of individual pulses, pulse frequency (pulses per second) and pulse duration as well as pulse pattern (burst, continuous or amplitude modulated). The definition of a TENS machine is any stimulating device that can deliver electrical currents across the intact skin surface. In theory, TENS is thought to selectively activate large-diameter Aβ nerve fibres without activating small-diameter nociceptive Aδ and C fibres or efferent fibres to muscles [479,480]. In clinical practice in humans, TENS is used to deliver an analgesic stimulus around a site of pain using pulse frequencies between 1 and 250 Hz and pulse durations between 10 and 1000 μs [479].

The clinical theory behind TENS is based on the neural gate control concept. The gate control theory is based on the fact that small diameter nerve fibres carry pain stimuli through a “gate mechanism” at the level of the spinal cord. Larger diameter nerve fibres carrying sensory (non-pain) signals going through the same gate can inhibit the transmission of the smaller nerves carrying the pain signal. Therefore, the pain gate may be shut by stimulating nerves responsible for carrying the sensory signal from peripheral mechanoreceptors. This also enables the relief of pain through massage techniques, rubbing, and the application of warm wheat bags and ice packs. The application of TENS is thought to allow the non-noxious sensory nerve signals stimulated to achieve preferential routing at the spinal cord, closing the “pain gate”.

The original research papers examining TENS identified by the systematic review criteria all involved human test subjects. Ward et al. [481] used a standardised cold water bath pain model in healthy young adults (*n* = 20) to examine the effects of TENS. Participants were required to immerse their hand and wrist in a cold-water ice bath at 0 °C and self-report their cold pain threshold. The measurement used was the duration in seconds until the threshold of a deep aching pain was reached. TENS was applied to the forearm at a pulse duration of 125 μs and was found to increase the pain tolerance (duration) of the cold-water bath immersion test by 25%, compared to the same subjects when the TENS was turned off. A similar approach was used by Shanahan et al. [482], who used the same cold-water bath pain model in healthy young adults (*n* = 20) to examine the effects of TENS. TENS was applied to the forearm in a biphasic pulse pattern (two 100 μs pulses) and at a frequency of 100 Hz. The researchers recorded self-reported duration to pain threshold as well as the perceived intensity of the pain on a visual analogue scale (VAS). TENS increased the duration to pain threshold and decreased pain intensity on the VAS. A criticism of both the studies by Shanahan et al. [482] (2006) and Ward et al. [481] (2009) is that it would have been obvious to the subjects when the stimulation was applied, and thus a placebo effect cannot be discounted. To be fair to these experimenters, their major hypotheses concerned a comparison of TENS with other electro-analgesia approaches (reviewed below).

In a truly clinical setting, Luchesa et al. [483] examined the effectiveness of TENS at providing analgesia for human patients (*n* = 30) following thoracic surgery for coronary artery bypass. Either TENS or a placebo electrical current (details not specified) were applied from 24 h after surgery for 5 days (2 daily sessions of 50 min each), with electrodes placed on the chest area. TENS was applied with a 125 μs pulse at 80 Hz. Patient reporting of pain levels (0–10 scale) were significantly improved by the TENS application compared with Control, although there were no consequent differences in respiratory function.

Percutaneous electrical nerve stimulation (PENS) is similar to TENS, except that needles are used to deliver the current through the skin, rather than via electrodes placed on the skin (in TENS). Ahmed et al. [484] examined the effectiveness of PENS in human patients (*n* = 30) suffering from recurrent headache of at least 6 months’ duration. PENS was applied via 10 fine needles placed into the skin of the scalp and neck and attached to a 15–30 HZ current with a pulse width of 500 μs. The control treatment was placement of the needles only. Both PENS and the control treatment were administered for 30 min, three times per week for 2 weeks, in a cross-over design with a 1 week stand-down period in-between. The results showed that the PENS treatment decreased VAS pain scores self-reported by the patients by approximately 55%, compared with a 20% reduction for the needles alone. The PENS treatment also improved patient physical activity and quality of sleep.

#### 3.8.2. Electroacupuncture (EAP)

Electroacupuncture is the electrical stimulation of precise acupuncture points in humans and animals to achieve the effects of acupuncture, including analgesia. Although there is still some debate on the mechanisms and range of efficacy of acupuncture, recent reviews cover a range of findings in humans and animals that support the hypothesis that endogenous opiates mediate the analgesic effects of acupuncture where such effects are present. Furthermore, studies using magnetic resonance imaging and positron emission tomography have detected physiological changes in the brain following peripheral acupuncture, although it cannot be determined if these changes are due to the mechanism of acupuncture or the alleviation of pain [485]. The application of EAP is also thought to provide analgesia via modification in the binding characteristics of serotonin receptors related to spinal antinociception. In addition, spinal glutamate receptors are activated in low frequency EAP analgesia in rats [463].

In contrast to the studies for TENS, the species breakdown for the following papers reviewed for EAP comprise studies with dogs (1), cats (1), rats (2), mice (4) and humans (1).

In mice, Chen et al. [486] examined the effects of EAP on vanilloid receptors in neurons, which are also implicated in the potential analgesic effects of capsaicin and its analogues (reviewed above). They found that EAP at specific acupoints attenuated TRPV1 and TRPV4 expression in mouse neurons following administration of either carrageenan or Freund’s Compete Adjuvant (FCA) to induce pain and inflammation. In reviewing this paper, it should be noted that the use of FCA in such a study is ethically questionable and almost certainly would not be permitted currently in Australia. Mouse behaviour tests (e.g., tolerance of hot surfaces) was also increased by EAP. Additionally, in mice, Lu et al. [487] found that EAP attenuated TRPV1 activation in mice injected with FCA. Building on these results, Xin et al. [488] reported that EAP in mice was able to induce both segmental (i.e., regional) and systemic analgesia tested by mechanical (pressure) and thermal pain thresholds, as measured by high-intensity EAP attenuating TRPV1, and low-intensity EAP mediating the activity of acid-sensing ion channel (ASIC)-3.

Yang et al. [489] used a FCA administration model in mice to measure the effects of EAP on the chronic pain that results. The findings showed that EAP reduced mechanical and thermal pain responses, probably mediated by TRPV1 effects. The FCA was administered to the hind paw, and the EAP point was adjoining the patella in the same leg. In a rat model, Du et al. [490] used FCA administration in the hind paw to show that EAP applied at an acupoint in the hind leg reduced expression of c-Jun N-terminal kinase in neurons of the spinal cord that had been upregulated following FCA, and that this effect was apparently independent of effects on TRPV1, suggesting multiple modalities for EAP effects. Additionally, using rats, Liu et al. [491] found that the EAP reduction in TRPV1 following FCA administration to the hind paw was mediated via a protein named Mas-related G-protein-coupled receptor C that is known to have a key role in modulating chronic inflammatory pain.

Using human subjects (*n* = 16 per treatment group), Montenegro et al. [492] used TENS-based stimulation of known acupoints (hence effectively using EAP) to measure effects on pain thresholds following hand immersion in an ice-water bath. A placebo (control) treatment was used, in which the electrode was applied, and the subjects told that they would be receiving an electrical current. The study results showed that EAP at the TE5 and PC6 acupoints (both on the forearm) increased the latency to pain threshold–i.e., had an analgesic effect.

In a full clinical setting using cats, Nascimento et al. [341] studied the effects of an EAP on cats (*n* = 10 per treatment group) undergoing ovariohysterectomy (spaying). An electrical stimulator was used to stimulate two acupoints (ST-36 and SP-6), a control group had no EAP treatment, and a third group received infrared laser stimulation of the same acupoints. A blinded observer used standardised pain scoring scales to rate the pain of the cats at intervals following surgery. If pain reached a defined threshold, then a pharmacological analgesic was administered—an important ethical consideration. The results showed that although overall pain scores did not differ between treatment groups, the control cats reached the defined threshold and received pharmacological analgesia more frequently that the cats stimulated at the two acupoints. In dogs, Cassu et al. [493] also studied the effectiveness of EAP following ovariohysterectomy. Animal numbers were low (*n* = 6 per treatment). The treatments were applied for 45 min following sedation, but before the induction of surgical anaesthesia. The treatments were (i) electrical stimulation of the skin adjoining the incision line; (ii) EAP of acupoints (ST-36, SP-6 and GB-34–i.e., similar to that used in the previous study with cats); and (iii) EAP and skin stimulation–a combination of i and ii. A blinded observer recorded pain scales at intervals following surgery, and ‘rescue analgesia’ was applied pharmacologically as needed (as per the previous study with cats). The results showed that dogs receiving treatment incorporating EAP had lower pain scores at 1 h after surgery.

#### 3.8.3. Interferential Current Analgesia (IFC)

Another variation that has been developed is that of interferential currents (IFC). The application of IFC aims to deliver currents to deep-seated tissue within the patient. Currents with a medium to high frequency (approximately 4000 Hz) are used in an effort to overcome skin impedance and penetrate more deeply. Some variations of IFC use two high frequency currents, slightly out of phase, to produce an amplitude-modulated wave of approximately 100 Hz [479]. Since its introduction in the 1950s in Austria, IFC has been used in human clinical practice for reducing pain and accompanying symptoms following musculo-skeletal injury. However, there have been limited studies assessing its efficacy.

Beatti et al. [494] used a study with human subjects to examine whether applied IFC actually penetrated and spread through deeper tissues as postulated. Using a small study with 12 human subjects, the experimenters measured the penetration of voltages into the thigh muscle following IFC application via surface electrodes. The results showed that sub-surface voltages were detectable, and that unsurprisingly voltage decreased with tissue depth. There was some voltage ‘spread’ beyond the area circumscribed by the electrodes. The authors concluded that their results gave some support to the concept of IFC, but that more research was needed. The study by Shanahan et al. [482] reviewed above in relation to TENS, also included an IFC treatment applied through the same skin electrodes (at a different time to the TENS). The results showed that the IFC was only marginally beneficial, and less effective than the TENS.

Equivocal results were also obtained in a study by McManus et al. [495], using healthy young human subjects (*n* = 20), in which the volunteers were subject to both thermal pain (ice-bath hand immersion) and mechanical pain (algometer pressure) on separate occasions, with and without concurrent IFC. The results showed that IFC delayed pain thresholds for both cold and mechanical sensation but did not alter the subjects’ reported experience of the mechanical pain.

#### 3.8.4. Transcutaneous Spinal Electroanalgesia (TSE)

Transcutaneous spinal electroanalgesia uses very short electrical pulses applied to the spinal cord in humans via self-adhesive electrodes placed on the skin in the appropriate area. Electrical pulses pass through the skin and underlying tissues to the spinal cord, causing a mild tingling sensation to the human patient. All the relevant TSE studies identified using the systemic criteria involved human subjects and failed to convincingly demonstrate analgesic effect.

Simpson et al. [496] studied the effects of TSE in human patients (*n* = 8) with chronic limb pain resulting from ischaemia due to inadequate vascular perfusion. The study design was a crossover incorporating a placebo treatment using an inactive TSE machine. The results showed no benefits of TSE for limb ischaemic pain in terms of self-reported pain levels, physical function, mood or sleep quality. Examining a different type of human pain presentation, Thompson et al. [497], examined the effects of TSE on lower back pain in 58 subjects, divided and allocated to each treatment, with either TSE or placebo treatments. The placebo treatments were administered by inactivated TSE machines, with this deactivation not externally apparent. No differences in pain score were recorded between the active and sham-treated participants. Palmer et al. [498] incorporated a TSE treatment in a study with healthy human subjects (*n* = 8 or 9 per treatment). The control treatment involved electrode placement with participants being told they would receive a form of TSE than could not be felt. In comparison with the control treatment, TSE produced no effects on pain responses, including mechanical pain measured by algometer.

#### 3.8.5. Other Electrical Modalities

Marchand et al. [499] investigated the analgesic and placebo effects of direct stimulation of the thalamus in the brain, in human patients (*n* = 6) with chronic pain problems, following thalamic stimulation electrodes being implanted by a neurosurgeon. The placebo treatment replicated the visuals of thalamic stimulation, without stimulation actually occurring. The patients were recorded over a 2 week period, both at home and during visits to the clinic. The researchers found that while thalamic stimulation improved reported pain scales, so did the placebo treatment, albeit to a lesser extent.

Gabis et al. [500] studied human patients with chronic back pain (*n* = 20 per treatment), allocated to either transcranial electrical stimulation or a placebo control. Transcranial electrical stimulation involves non-invasive electrodes being placed on the head. The control treatment involved an inactive stimulation machine that was blinded to both patient and operator. The results showed that transcranial electrical stimulation increased circulating beta-endorphin concentrations, but that both the active and placebo treatments significantly decreased self-reported pain levels. 

Finally, an alternative to TENS, termed burst-modulated medium-frequency alternating current (BMAC), was used in the study by Ward et al. [481], reviewed in the TENS section earlier. The application of BMAC via the same electrodes (on a different occasion) showed that BMAC was as effective as TENS in increasing the latency to cold pain threshold. This result is interesting because functionally BMAC is very similar to IFC, which elsewhere has not yielded very promising results in objective, controlled studies.

#### 3.8.6. Photomodulation

Light therapy, otherwise known as photomodulation, has been shown to affect a number of biologic processes including activation of opioid receptors and the L-arginine/nitric oxide pathway [501] (both of which are involved in pain modulation and perception). Light in the visible and near infrared (NIR) portions of the spectrum have been evaluated.

Red coloured light-emitting diodes (LEDs) of 660 nm wavelength were applied directly to the paw of mice (*n* = 8–10) prior to injection of formalin to induce inflammation and pain [502]. When the LEDs were applied 60 min prior to the insult, there was a significant reduction in both the response to injection (acute pain) and a significant reduction in pain sensitivity during the delayed (inflammatory) response to the insult, measured 15–30 min post-injection. LEDs applied 30 s and 90 s prior to injection also reduced pain sensitivity during the delayed response but had no effect on acute pain sensitivity. Similarly, in a plantar incisional model of pain, mice (*n* = 8) first received a longitudinal incision (under general anaesthesia) to the hind paw, followed by infrared LED therapy using a 950 nm wavelength applied to the incision once daily for 5 days, starting the day after incision [501]. After a single LED treatment, the nociceptive response was significantly reduced for at least 1 h following treatment, while daily treatment led to significant reductions in nociceptive response on days 1, 2, 3 and 4 following plantar incision, as compared with non-LED-treated controls. By day 5, the nociceptive response of the controls had returned close to sham controls. In a sciatic nerve crushing model, Cidral-Filho et al. [503] demonstrated that application of 950 nm infrared LED light, daily for 15 days, to the skin over the crush injury reduced tumour necrosis factor alpha (an inflammatory cytokine) and reduced the nociceptive response in mice (*n* = 8), suggesting that LED therapy could have potential use in treating neuropathic pain. Pigatto et al. [504] also showed that infrared LEDs (wavelength 890 nm), applied daily for 20 min, reduced the nociceptive response in mice (*n* = 8) undergoing either sciatic nerve injury or ischaemia-reperfusion injury (both of which are models of chronic pain).

Light can be polarized, meaning that the waveform has been aligned such that the oscillation of the wave is perpendicular to the direction of travel (without polarisation, the waveform is much more random). Polarized light applied to the skin over the trigeminal nerve (on the cheek) reduced pain sensitivity both on the treated skin and on the forearm [45], suggesting that the effects of photomodulation of one nerve can be transferred to other nearby neural pathways. Muneshige et al. [61] used polarized near-infrared light (wavelength range 600–1600 nm) allied to the lumbar area of rats (*n* = 11) in a sciatic nerve ligation model. A single treatment of 60 s irradiation significantly reduced hyperalgesia for at least 8 h post-treatment, while daily application, beginning one week after nerve ligation significantly reduced hyperalgesia for up to one month (after which the non-irradiated controls were returning to normal. In humans, patients with chronic arthritic or myofascial pain received polarized light (830 nm) over the ‘pain points’ each week for three weeks as an adjunct to local anaesthesia and glucocorticoid therapy [505]. Patients receiving the light treatment reported a greater reduction in pain that patients receiving only local anaesthesia and glucocorticoids, with the light apparatus applied to the ‘pain point’ but not switched on.

Laser is amplified light at a single wavelength, resulting in a greater intensity of energy than achieved using incandescent or LED approaches. It may or may not be polarized. Clinical use of laser therapy to reduce pain is termed Low Level Laser Therapy (LLLT). LLLT has effects on nerve conductance and modulates the endogenous opioid system [50,506]. In humans, both LED and LLLT have resulted in lower pain scores in the week following coronary bypass surgery compared to placebo controls [507]. Patients (*n* = 30 per group) received LLLT at 640 nm or LED at 660 nm immediately following surgery and then every second day till day 8. Similarly, patients (*n* = 28–30) undergoing dental realignment using orthodontic separators also reported significantly less pain after receiving LLLT (635 nm) or LED (635 nm) every 12 h for 1 week post-insertion of the separators, compared to control patients receiving no irradiation [508].

#### 3.8.7. Electromagnetic Field Therapy

Electromagnetic fields (EMFs) may be static (e.g., the geomagnetic field or from static magnets) or time-varying (e.g., thunderstorms or the electromagnetic spectrum). The electromagnetic spectrum includes light (reviewed above), radio-frequency, microwave, x-ray, ultra-violet, infrared or gamma waves. EMFs can affect a variety of biological systems, including pain perception and modulation pathways [509,510,511,512,513,514]. Interestingly, EMFs can either increase nociceptive sensitivity (induce hyperalgesia) or decrease nociceptive sensitivity (induce analgesia), and this translates to the impact on concurrent medication: EMF having been found in separate studies either to enhance or to suppress the analgesic effects of opioid drugs [510,511,515]. It is likely that the parameters of the EMF alter the outcome, but it appears that the optimal parameters to consistently achieve analgesia are yet to be fully defined.

Pulsed electromagnetic field (PEMF) does appear to show promise as a therapy. Shafford et al. [516] used PEMF postoperatively in dogs (*n* = 4) undergoing ovariohysterectomy (spaying) and demonstrated a non-significant reduction in pain score in the first three hours post-operatively as compared to control dogs. A further group of dogs that received morphine as well as PEMF showed significantly reduced pain scores at 30 min post-operatively as compared to control dogs. In humans, PEMF has been assessed for treatment of persistent back pain, with 33% of patients (*n* = 34) reporting an improvement in pain score and mobility [517]; and for treatment of postoperative pain in cosmetic surgery (*n* = 14), with PEMF-treated patients reporting significantly lower pain scores than untreated controls [518]. In both studies, PEMF was delivered repeatedly: for back pain, PEMF treatment was 30 min twice daily for 45 days; for post-operative analgesia, PEMF was delivered for 30 min every 4 h for the first 3 days, every 8 h for the subsequent 3 days, then twice daily till day 8 post-procedure. Although automatic PEMF devices are available in human medical practice, this treatment regime is rather impractical for a livestock husbandry setting.

Transcranial magnetic stimulation (TMS) is used to treat schizophrenia and other neurological conditions. A changing magnetic field on the scalp induces electrical currents in the brain. Ambriz-Tututi et al. [27] assessed the effects of TMS on nociceptive threshold in rats (*n* = 6). Rats received 60 Hz TMS for 2 h twice daily for 14 days. There was a significant reduction in nociceptive threshold (allodynia) in TMS-treated rats compared to control rats. Furthermore, when formalin was injected into the hind paw, TMS rats demonstrated hyperalgesia as compared with non-TMS formalin-injected rats.

#### 3.8.8. Cryoanalgesia

Cryoanalgesia–sometimes specifically termed ‘cryoneurolysis’–refers to the application of a low temperature to a nerve that reversibly curtails signal transmission, and thus may provide analgesia. Such cryoanalgesia is often delivered via a cryoprobe–a hollow cannula containing an even smaller cannula through which is passed a cryogen gas at high pressure. The cannula tip is closed, but an ice-ball forms within the tip, extracting heat from the surrounding tissues. Unlike freeze-branding of animals (e.g., for identification purposes), in which the intent is to use the cold application to kill the tissues, cryoanalgesia aims to affect neural transmission while maintaining tissue integrity. The majority of the papers identified in the systematic review process concerned human patients, with one experimental study examining effects in healthy human subjects, and two animal-based experimental studies.

Ju et al. [519] examined one of the potential side-effects of repeated cryoanalgesia—a form of hyperalgesia presenting as neuropathic pain in human patients. This has been hypothesised to be due to repeated, chronic nerve damage due to the multiple applications of cryoneurolysis. In the study, rats had their sciatic nerve dissected to exposure, and then lesioned with a 60 s freeze, or not frozen (controls). Following this severe cryoneurolysis, the mechanical pain threshold of the treated rats indicated hyperalgesia, and this was accompanied by increased expression of nerve growth factor, which the researcher hypothesised was playing a role in the hyperalgesia. Similarly, Hsu et al. [520] examined histological changes in rat sciatic nerves following repeated cryoneurolysis. Rats (*n* = 30 per treatment) were either untreated controls, received one administration of cryoneurolysis, or received 3 cryoneurolysis treatments at 6 week intervals. Histological analysis of the nerves undertaken at intervals thereafter showed that cryoneurolysis caused axonal damage followed by regeneration by 24 weeks post-treatment. Physiological function was recovered by 8 weeks post-treatment.

Much of the interest in cryoanalgesia in human medicine is centred around its potential use post-thoracotomy. Ba et al. [521] compared the use of the NSAID parecoxib with cryoanalgesia via cryoprobe of the intercostal nerves for thoracotomy for lung surgery in human patients (*n* = 89 per treatment). The self-reported pain scores and morphine use was less in patients following cryoanalgesia. Similarly, Gwak et al. [522] examined the use of cryoanalgesia of the intercostal nerves for thoracotomy in combination with intravenous continuous analgesia incorporating the opioid fentanyl. All patients received intravenous analgesia, and some also had cryoanalgesia (*n* = 25 per treatment group). The inclusion of the intercostal cryoanalgesia did not affect pain scores but did result in some improvement in respiratory function on the 7th day post-surgery (but not on other days). Ju et al. [523] compared intercostal nerve cryoanalgesia with epidural bupivacaine and morphine for control of pain post-thoracotomy (*n* = 53 or 54 per treatment group). There were no differences in self-reported pain scores for the first 3 days post-operatively, but there was some neuropathic pain in the cryoanalgesia group in the months that followed. Morikawa et al. [524] examined the use of intercostal nerve cryoanalgesia (*n* = 6 patients) supplementing the use of standard pharmacological pain relief post-thoracotomy (*n* = 13 patients with no cryoanalgesia). Pain scores were not different, but the patients supplemented with intercostal nerve cryoanalgesia had a lower opioid requirement and were discharged on average 1 day earlier.

Bellini and Barbieri [525,526] reported data from a series of cases with human patients suffering from chronic pain including sacroiliac pain and knee pain. The use of a cryoprobe to achieve cryoneurolysis resulted in a self-reported improvement in VAS pain scores by one month, with optimum results at 3 months. Cavazos et al. [527] examined the effectiveness of cryoneurolysis for the treatment of chronic heel pain in human patients. Using surgical dissection to aid cryoprobe placement and comparing self-reported pain scales after surgery on 137 feet, the results showed that 77% of procedures were successful as measured by substantial improvement in pain scales. Coelho et al. [528] studied the effectiveness of cryoanalgesia as a supplement to topical local anaesthetic for cataract eye surgery in 25 patients (one eye operated with supplemental cryoanalgesia and the other without). Cryoanalgesia was performed using eye irrigation with a cold solution (4 °C), or at room temperature (controls). The results showed that there was no difference in pain scores.

Finally, studies with human patients have examined the effectiveness of cryoanalgesia in relation to a variety of procedures, including tonsillectomy, arterial puncture, amniocentesis, and in the relief of occipital nerve pain. Kim et al. [529] examined the use of cryoneurolysis for the treatment of occipital nerve pain in conjunction with local anaesthetic infiltration. However, because the use of cryoanalgesia was only applied for patients with more severe symptoms, the results are not conclusive. Robinson and Purdie [530] conducted a randomised controlled trial incorporating supercooling of the tonsillar fossa to between −20 and −32 °C for 1 min immediately following tonsil removal, compared with no cryoanalgesia (*n* = 29 or 30 per treatment). Patients that had received cryoanalgesia had a reduction in self-reported pain scores in the 10 days post-surgery and returned to work or school on average 4 days earlier.

More familiar may be the application of ice packs or coolant sprays to the skin. Working with healthy human volunteers (*n* = 60), Al Shahwan [531] implied that the pain of local anaesthetic infiltration may be lessened by ice treatment of the skin beforehand, but unfortunately the study was confounded in that the treatments for comparison were (unbuffered local anaesthetic with no pre-icing) versus (unbuffered local anaesthetic following pre-icing). Haynes [532] reported a randomised controlled trial in which patients (*n* = 40 per treatment group) had either a bag of ice placed on their forearm for 3 min before arterial blood sampling, or no pre-treatment. Self-reported pain scores were lower with ice pre-treatment (noting that no placebo was possible), but interestingly this effect was only significant in patients undergoing the procedure for the first time. Similarly, Hanprasertpong et al. [533] examined the application of an ice pack on the skin before needle amniocentesis in women (*n* = 184 or 188 per treatment group). Again, no placebo was possible, but the women receiving the ice pack pre-treatment reported moderately lower pain scores post-procedure. In 2–4 month-old calves (*n* = 10), a topical vapocoolant spray was applied to the scrotum prior to incision and then to the spermatic cords after exteriorisation of the testes for castration, but there was no evidence of an analgesic effect, based on post-procedural behaviour scoring and infrared thermography of the eye [232].

#### 3.8.9. Section Summary

Although PEMF, EAP or TENS can assist in reducing post-surgical pain, application for a prolonged period pre and/or post-procedure is required, which is not practical under commercial livestock production conditions. Topical ice or vapocoolant may assist in reducing the pain response as part of a multi-modal analgesic approach, but this is yet to be confirmed.

### 3.9. Delivery Systems

The key to optimising efficacy of any analgesic is optimising the delivery of the agent to the target tissue receptor(s). From a purely pharmacological point of view, this could mean improving the bioavailability of the agent (how much is absorbed), reducing the excretion of the agent (so that circulating levels are maintained for longer), or in the case of centrally-acting agents, improving the ability of the agent to cross the blood–brain barrier and reach the target receptor(s). Nano-encapsulation of tizanidine (a centrally active anti-spasmodic agent) and nano-emulsion of tramadol improved their absorption across the nasal mucosa and transport across the blood–brain barrier [393,534], while co-crystallisation of an agent can improve its water-solubility and thereby improve absorption [429]. Microparticles or micro-beads impregnated with an agent can provide a sustained-release approach to administration, extending the duration of action of agents including local anaesthetics and NSAIDs [178,360,361,535,536]. For example, a nerve block using bupivacaine-impregnated microspheres in a dexamethasone solution provided analgesia for 21 h in rats, as compared to 2 h using bupivacaine solution alone [241].

From a practical point of view, development of delivery systems that are consistent, safe and easy to apply in the field are essential for any analgesic agent to be used in livestock production. Needle-less (e.g., PULSE NeedleFree^®^, Pulse NeedleFree Systems Inc., Lenexa, KS, USA; PharmaJet^®^, Golden, CO, USA) approaches or guarded needle tools (e.g., NUMNUTS^®^, Senesino Ltd., Glasgow, UK. Sekurus^®^ injectors, Simcro, Hamilton, New Zealand) for injectables are gaining popularity in livestock production as a safer alternative to unguarded needles and syringes. Formulations that allow for oral or oral transmucosal administration of agents are entering the marketplace (e.g., Ilium Buccalgesic OTM, Troy Laboratories, Sydney, Australia), and transdermal approaches are also under investigation, while an in-feed formulation of an NSAID may provide sustained post-procedural analgesia [370]. Continued development of practical delivery systems and formulations is to be recommended.

## 4. Conclusions

It is evident from the enormous volume of research underway that the ‘pain and analgesia’ challenge has not been solved for any species, including our own. Therefore, it is unreasonable to expect that we will solve this challenge for the livestock species in the near future, we can merely begin to alleviate pain in order to improve animals’ ability to thrive. This review merely scratches the surface of the available literature, and future research into any of the areas described will require focused literature evaluation to eliminate duplication and identify the most promising approaches to develop new analgesic strategies for livestock.

Detailed understanding of the physiology of pain perception, particularly at the molecular level, is still developing. There is increasing evidence that the pain perception pathways continue to develop after birth [537], and events in the early neonatal period can affect subsequent pain sensitivity. A number of non-pharmacological factors can affect the pain response, and these warrant further investigation toward the development of an holistic approach to integrated pain management.

Major challenges remain in assessment of pain in a commercial setting. Further development of novel measures of pain that can be automated through using new (e.g., sensor) technologies to overcome the issues with practical application in the field is warranted.

The complex processes of induction and central processing, and consequent adverse outcomes suggest that measurement of pain will continue to rely on a combination of measures. A focus on developing a single unified measure is probably inappropriate because pain is multidimensional in terms of the experience and the consequences.

Of the NSAIDs, meloxicam does appear to have the longest elimination half-life, and therefore is the most practical agent for use in livestock husbandry, with flunixin a close second. However, there are indications that carprofen can persist in circulation, although pharmacokinetic data are not reported. To date, efficacy studies with carprofen have shown no particular benefit, but the dose rates used have not been based on pharmacodynamic data and may be below the minimum effective concentration to provide effective analgesia. Further work on NSAIDs should begin with characterisation of the pharmacokinetic and pharmacodynamic properties of an agent, and establishment of an estimated minimum effective concentration prior to conducting efficacy studies. Large differences exist between species in terms of the metabolism and pharmacokinetics of any particular agent, such that extrapolation of dose rates is not appropriate and may even be dangerous to the animals concerned [367].

Use of local anaesthesia with NSAIDs, for livestock undergoing routine husbandry procedures does provide greater amelioration of the pain response than use of a single agent alone and should be recommended as current best practice. It may be of value to industry, supporting adoption, to continue to systematically evaluate multi-modal approaches to the various husbandry procedure methodologies (e.g., surgical or ischaemic) and combinations (e.g., mulesing with castration).

The duration of analgesic effect provided by local anaesthetics can be extended by the addition of various compounds, for example α-agonists (clonidine), neostigmine, sodium-channel blockers. Regulatory aspects around some of these compounds may limit the practicality of such formulations for in-field use, but if safety and efficacy can be demonstrated, such formulations may well have a place.

The ion channels and molecular receptors recently identified show promise as targets for new analgesic agents. For example, TRPV1 has been associated with a variety of pain conditions including constriction type nerve injury, neuralgia and inflammatory conditions, in which case TRPV1 antagonists may be of benefit post-procedures such as ring castration and tail docking, in which there is constriction, nerve damage and subsequent inflammation as the tissue necroses and sloughs. However, development of these agents is still predominantly in its infancy, and although research can be carried out using individual chemicals prepared in the laboratory, it will be a number of years before a formulation can be made available for the livestock industries. From initial formulation, the regulatory process requires safety, toxicity, tissue residue and efficacy studies prior to registration of the formulation. The ‘natural’ ion channel activators camphor, eugenol and capsaicin are closest to market, being used in human preparations. Camphor and eugenol (oil of cloves) may be of interest, particularly to organic growers, as topical treatments for ear tagging or shearing cuts, but analgesic efficacy in these situations is yet to be demonstrated.

Delivery systems that are consistent, safe and easy to apply in the field are essential for any analgesic agent to be used in livestock production. Continued development of such systems, and formulations that allow for sustained analgesia (e.g., sustained-release formulations, in-feed medication) is to be recommended.

Much of the recent research into pain and analgesia focuses on the treatment of chronic pain conditions, e.g., neuropathic pain. In light of the finding that cautery tail docking, in piglets, can lead to sustained changes in the spinal tissues, there would be value in exploring the potential for these novel analgesics to prevent the development of these changes. If that can be achieved, such agents could provide a valuable addition to a multi-modal approach to management of pain associated with husbandry procedures.

One aspect for which data are glaringly absent is the enterprise-level benefit of use of analgesia for routine husbandry procedures. In many studies, a reduction in feeding behaviour and growth rate in the first few days post-procedure is reported, but a compensatory increase in feeding behaviour and growth over subsequent days leads to no significant differences in growth being observed between treatments after 2–3 weeks. However, these studies are all small-scale, and tend to measure solely bodyweight as the production parameter. There may be effects on, for example, feed conversion efficiency; longer-term growth; immune competence; resilience to challenge that could result in morbidity or mortality. In light of the myriad factors affecting livestock production parameters (including, but not limited to, genetics, feed availability, feed quality, parasitism, weather/climate), any study aiming to identify such effects must be very-large-scale (similar to the Phase III clinical trials phase when bringing an entirely new drug or vaccine to market) and extremely well controlled. In this respect, the increasing popularity of feedlotting of lambs may provide an opportunity to begin to gather production-parameter data, and simulation modelling may assist in generating predictions of potential benefit.

## 5. Limitations of This Review

The scope of this review included indexed peer-reviewed scientific literature published between 2000 and September 2019. Other data may exist in earlier literature or in the form of non-peer-reviewed documents, for example patents, reports submitted to a pharmaceutical registration authority, or industry reports. Furthermore, the overwhelming quantity of published articles addressing analgesia in this period requires an almost brutal preliminary screening and exclusion process in order to reduce the volume of literature to a manageable level. Much more information, pertaining in particular to the detailed physiology of pain and analgesia, is present in the scientific literature.

It very quickly became clear that there is a distinct lack of consistency between studies, such that systematic comparison of research outcomes is incredibly difficult. Even within species and pharmaceutical agents, differences in age of animal; dose rate; route of administration; pain model (or procedure assessed) and measures used confound comparison of outcomes. A general impression was that there appeared to be a lot of assumptions underpinning many studies: assumptions that the dose in one species can translate to another species; assumptions that the minimum effective concentration in one species also can be transferred to another species; assumptions that the behavioural response to an insult might be the same across vaguely similar insults.

## Figures and Tables

**Table 1 animals-11-01127-t001:** Literature Search Criteria.

Databases to Search	Web of Science Core Collection PubMed MedLine Scopus
Inclusions	Research papers Analgesic agents Husbandry procedures Livestock Companion animals Humans
Exclusions	Policy documents Reviews (except as a means to identify other research) Philosophical/opinion papers Patents General anaesthesia Papers published prior to 2000 * Language other than English

* Although the focus of this review is on papers published between 2000 and September 2019, some earlier papers have been cited in this review in order to provide context.

**Table 2 animals-11-01127-t002:** Agreed Keywords for Literature Search.

Category or Primary Keywords	Secondary or Qualifier Keywords
Livestock	Sheep; cattle; goat; lamb; calf/calves; kid; ewe; wether; cow
Companion animals	Horse; cat; dog; pet; rabbit; rat; mouse
Analgesic	Pain; discomfort; NSAID; opio *; * cannabi *; local an * sthetic; sedative
Pain receptor	COX; ASIC; sensation; * algesia; neurophysiology
Husbandry procedures	Surg *; mules *; castrat *; dock; amputate *; dehorning; disbud; lame *; shear *; injur *

* indicates variations in spelling or suffix. NSAID: non-steroidal anti-inflammatory drugs. COX: cyclooxygenase, ASICs: Acid-sensing Ion Channels.

**Table 3 animals-11-01127-t003:** Information Collated into Stocktake Catalogue.

Field Title	Description or Example of Content
First author	Surname
Year	Year of publication
Country	Country in which study was carried out
Affiliation	Research organisation involved
Study category	Pharmacokinetics, behaviour, physiology, etc.
Species	Species used during study
Sample size	Number of animals in treatment group
Animal type	Dairy, beef, mature, neonate, etc.
Procedure	Pain model used, if applicable., e.g., mulesing and castration
Agent	Drug evaluated
Measure	Specific physiological assay; specific behaviour/posture
Finding	Significant/non-significant or other comment
Key conclusions	Main points from article conclusion
Notes/comments	Free-form comments section
Article reference	Citation information

**Table 4 animals-11-01127-t004:** Mean half-life of elimination and minimum effective concentration (MEC) in plasma of various NSAIDs in different species, reported in the period 2000–2019.

Agent (Route of Administration)	Parameter	Sheep	Goats	Cattle	Horses	Other
Acetaminophen (intravenous)	Half-life			1.28–1.73 h [171]		
	MEC *					
Carprofen (intravenous)	Half-life				0.25–0.95 h [172]	
	MEC *				<0.7 mg/kg	
Carprofen (intramuscular)	Half-life					White Rhino: 105.71 [173]
	MEC *					
Diclofenac (intravenous)	Half-life	2.84 h [174]				
	MEC *					
Diclofenac (intramuscular)	Half-life	2.12 h [174]				
	MEC *					
Flunixin meglumine (intravenous)	Half-life			6 h [175,176]		
	MEC *					
Flunixin meglumine (in feed)	Half-life	7.95 h [177]				
	MEC *					
Ibuprofen (BeadBlock embolization)	Half-life	2.76 h [178]				
	MEC *					
Ketoprofen (intravenous)	Half-life	0.63 h [179]	0.18–0.19 h [180]	0.71–0.91 h (adult) 1.35–1.7 h (neonate) [181]	0.13–2.67 h [182,183]	Cat: 1.52 h [184] Buffalo: 3.58 h [185]
	MEC *	<1.5 mg/kg	<3 mg/kg			Cat: <2 mg/kg [184]
Ketoprofen (oral)	Half-life					Cat: 0.57–0.92 h [184]
	MEC *					Cat: <2 mg/kg [184]
Ketorolac (intravenous)	Half-life	0.30 h [186]				
	MEC *					
Ketorolac (intramuscular)	Half-life	0.25 h [186]				
	MEC *					
Mefenamic acid (intraperitoneal)	Half-life					
	MEC *					Mice: <1.5 mg/kg [187]
Meloxicam (intramuscular)	Half-life	12.6–12.8 h [188]	10.82 h [189]			
	MEC *					
Meloxicam (intravenous)	Half-life	10.85–14.0 h [190,191]	6.07–12.73 h [189,192]	8.1–21.86 h [193,194,195]		Iguana: 9.93 h [196] Llama: 17.4 h [197] Pig: 6.15 h [198] Buffalo: 12.4 h [193] Donkeys: 0.97–1.02 h [199]
	MEC *					
Meloxicam (oral)	Half-life	15.4 h [191]	10.69 h [190,192]	14.59–27.54 h [175,176,195,200,201]	10–34 h [202]	Iguana: 12.96 h [196] Llama: 22.7 h [197] Pig: 6.83 h [198]
	MEC *					
Meloxicam (subcutaneous)	Half-life	10.8–14.2 h [188]	15.16 h [192]	16.2 h [200]		Cat: 37 h [203]
	MEC *					
Sodium salicylate (intravenous)	Half-life	0.46–1.16 h [204]		0.62–0.68 h [205,206,207]		
	MEC *					
Sodium salicylate (oral)	Half-life	1.86–1.9 h [204]				
	MEC *					
Tepoxalin # (intravenous)	Half-life					Broilers: 1.07 h [208]
	MEC *					
Tepoxalin (oral)	Half-life				6.3–8.8 h [209]	Broilers: 2.8 h [208] Rabbits: 2.8–3 h [210]
	MEC *				<10 mg/kg [209]	Broilers: >30 mg/kg [211]
Tolfenamic acid (intramuscular)	Half-life		2.29–2.61 h [212]	6.68–8.22 h [213]		
	MEC *					

* MEC is based on the minimum dose rate at which suppression of prostaglandin or thromboxane production has been reported. # Data presented relate to RWJ-20142, the active acid metabolite of tepoxalin. NSAID: non-steroidal anti-inflammatory drugs.

## Data Availability

All data are available in the references cited.

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
