# Peer review of "Analgesia for Sheep in Commercial Production: Where to Next?"

_animals, 2021, doi:10.3390/ani11041127_

Round 1

Reviewer 1 Report

This paper will be extremely useful to researchers and policy makers in the area of farm animal production/welfare as a comprehensive review of the pain literature from 2000-2019.

I found the headings a little confusing, in that the Section summaries should all be at the same level of numbering. It may require some "subsection summaries"

There are a few typos, which is not surprising in a review of this length, these are the type of mistakes that will not be caught by spell check, so please read through carefully.

e.g. line 272 -- I think you mean "weight"

line 1390 -- I think you mean "nerve"

Author Response

Dear reviewer, thank-you for your comments, which have been addressed as follows:

I found the headings a little confusing, in that the Section summaries should all be at the same level of numbering. It may require some "subsection summaries"

We agree – we have re-named the ‘sub-section’ summaries accordingly.

There are a few typos, which is not surprising in a review of this length, these are the type of mistakes that will not be caught by spell check, so please read through carefully.

We have done so, and have corrected a number of errors.

e.g. line 272 -- I think you mean "weight"

This has been corrected

line 1390 -- I think you mean "nerve"

This has been corrected

Reviewer 2 Report

Overall the quality of the paper is high.  I do think, however, that there is unnecessary detail about some areas which is beyond that indicated in the title of the paper.  For example the detail of some anesthetic additives (morphine), the detail of the biochemistry and physiology of COX systems, detailed pain receptor physiology, TENS, electro acupuncture and IFC details would seem too detailed for a paper on commercial sheep applications.

I would have liked to see more detail of comparative pain assessment methodologies and their comparisons. Cortisol and EEG systems for example which while mentioned were not analysed or evaluated. 

A few typos and a large passage without references are notated with comments in the attached file.

Author Response

Dear reviewer, thank you for your comments, which improve the manuscript. These have been addressed as follows:

I do think, however, that there is unnecessary detail about some areas which is beyond that indicated in the title of the paper.  For example the detail of some anesthetic additives (morphine), the detail of the biochemistry and physiology of COX systems, detailed pain receptor physiology, TENS, electro acupuncture and IFC details would seem too detailed for a paper on commercial sheep applications.

Perhaps we should agree to disagree – although the paper considers these agents in the context of commercial sheep applications, the review was written to be a reasonably comprehensive introduction to the topic area, pitched at regulatory, policy and strategy personnel as opposed to producers.

I would have liked to see more detail of comparative pain assessment methodologies and their comparisons. Cortisol and EEG systems for example which while mentioned were not analysed or evaluated. 

We felt that a detailed evaluation of comparative pain assessment methodologies was beyond the scope of this particular review – however, we have added the following to the sections on EEG: “Although these show promise for research purposes, they are likely to be more useful in small-scale controlled research applications that in large-scale commercial monitoring” and cortisol biomarkers: “Cortisol has been widely used in many studies, however care must be taken over interpretation, as a cortisol response can be elicited by any form of arousal, not merely pain.”

A few typos and a large passage without references are notated with comments in the attached file.

Line 108 – why not other infections

Other infections are likely to produce nerve damage through compression and trauma as opposed to intrinsic changes in the nerve cell. However, for completeness we have included ‘and other’ after ‘viral’.

Line 156 – this whole section has no references??

Lines 156 to 169 are comments of the authors and introduce the subsequent section

Line 186 – ischaemic pain?

Ah yes, our apologies for the inversion here – this has been corrected.

Line 262 – has?

We have corrected the grammar accordingly

Line 314 – does this need to be defined

We have added a definition of ‘grimace’

Line 618 – wording?

We have corrected ‘would’ to read ‘wound’

Line 624 – does this need to be defined

We have added a definition of Nav

Reviewer 3 Report

The document is a very complete Literature Review regarding analgesia in most of domestic mammals, it is suggested either to change the title (including Domestic mammals rather than Sheep only) or to select items and topics exclusive to sheep species. 

Some suggestions are included in attached file

Author Response

Dear reviewer, thank-you for your comments, which have been addressed as follows:

it is suggested either to change the title (including Domestic mammals rather than Sheep only) or to select items and topics exclusive to sheep species. 

This comment is also appended to Line 1448.

Perhaps we should agree to disagree – if we restrict the scope to items exclusive to sheep species, we limit our ability to scan the landscape for future opportunities, which was the main objective of the review. Using keywords such as ‘livestock’, ‘pain’ and ‘analgesia’ a reader interested in the broader topic applied to other domestic species will still be able to access the article.

Some suggestions are included in attached file

Line 68 – this table could be described in a paragraph.

Indeed it could – however we felt that there was quite enough text, and a table would provide a succinct representation and a rest for the eyes.

Line 448 – highlighted text

We assume this is highlighted because is a citation that appears in text format – the numbered citation of this article appears in line 153, after we have described the findings

It is suggested to reduce Conclusions, including only the most important points to highlight

Indeed, we thought long and hard on this – a short list of dot points would be very nice, but we felt that as the review itself was so large we needed to summarise the key points to help the reader and to close the discussion properly.